# EVENT2VEC: PROCESSING NEUROMORPHIC EVENTS DIRECTLY BY REPRESENTATIONS IN VECTOR SPACE

## ABSTRACT

Neuromorphic event cameras possess superior temporal resolution, power effi-
ciency, and dynamic range compared to traditional cameras. However, their asyn-
chronous and sparse data format poses a significant challenge for conventional
deep learning methods. Existing solutions to this incompatibility often sacrifice
temporal resolution, require extensive pre-processing, and do not fully leverage
GPU acceleration. Inspired by word-to-vector models, we draw an analogy be-
tween words and events to introduce event2vec, a novel representation that al-
lows neural networks to process events directly. This approach is fully com-
patible with the parallel processing and self-supervised learning capabilities of
Transformer architectures. We demonstrate the effectiveness of event2vec on
the DVS Gesture, ASL-DVS, and DVS-Lip benchmarks. A comprehensive ab-
lation study further analyzes our method's features and contrasts them with ex-
isting representations. The experimental results show that event2vec is remark-
ably parameter-efficient, has high throughput, and can achieve high accuracy even
with an extremely low number of events. Beyond its performance, the most sig-
nificant contribution of event2vec is a new paradigm that enables neural networks
to process event streams as if they were natural language. This paradigm shift
paves the way for the native integration of event cameras with large language
models and multimodal models. Code, model, and training logs are provided in
`https://anonymous.4open.science/r/event2vec_iclr-7B40`.

## 1 INTRODUCTION

Neuromorphic computing is an emerging research field that seeks to develop the next generation
of artificial intelligence by emulating the brain's principles (Mead, 1990). A significant advance-
ment stemming from this paradigm is the event camera, a sensor inspired by the biological retina
(Gallego et al., 2022). Prominent examples include the Dynamic Vision Sensor (DVS) (Lichtsteiner
et al., 2008) and the Asynchronous Time-based Image Sensor (ATIS) (Posch et al., 2011). Unlike
traditional cameras that capture synchronous frames, event cameras operate asynchronously, gen-
erating events in response to per-pixel brightness changes. This operational principle endows them
with exceptionally high temporal resolution (on the order of microseconds), low power consump-
tion, and a High Dynamic Range (HDR) exceeding 120 dB. This asynchronous operation results
in a sparse stream of events, typically encoded in the Address-Event Representation (AER) format.
An event is represented as a tuple $(x, y, t, p)$, composed of the pixel's spatial coordinates $(x, y)$, a
timestamp $t$, and a binary polarity $p$ that indicates the direction of the brightness change.

Most contemporary deep learning models are designed to operate on dense, regularly structured,
multi-dimensional tensors. This tensor-based paradigm is foundational to mainstream deep learning
(LeCun et al., 2015) and is ubiquitously employed in modern scientific computing and machine
learning frameworks, including NumPy (Harris et al., 2020a), TensorFlow (Abadi et al., 2016),
and PyTorch (Paszke et al., 2019). Consequently, the sparse and asynchronous nature of event
streams in the AER format is fundamentally incompatible with these tensor-based methods. To
address this disparity, substantial research efforts have been devoted to converting events to dense
representations, or designing new data and network structures to process the irregular events directly.

Existing methods primarily address the challenge of event encoding: how to effectively extract infor-
mation from events and represent it for processing by neural networks. This challenge is analogous

| Word | how | are | you |
|---|---|---|---|
| Index | 5269 | 527 | 499 |
| Position | 0 | 1 | 2 |

| Event | $E[0]$ | $E[1]$ | $E[2]$ |
|---|---|---|---|
| Index | $x[0], y[0], p[0]$ | $x[1], y[1], p[1]$ | $x[2], y[2], p[2]$ |
| Position | $t[0]$ | $t[1]$ | $t[2]$ |

Figure 1: Conceptual analogy between words and events. The illustration of the DVS 128 camera is adapted from Lichtsteiner et al. (2008).

to word encoding in natural language processing, a problem successfully addressed by foundational techniques such as word-to-vector (word2vec) (Mikolov et al., 2013). The word2vec model embeds each word into a fixed-length vector, enabling the relationships between words to be represented by mathematical operations between vectors. This vector representation approach is highly compatible with deep learning architectures and has become a foundational component of modern Natural Language Processing (NLP) models (Devlin et al., 2019a; Brown et al., 2020). We identify numerous parallels between words and events, as illustrated in Figure 1. The key similarities are as follows:

(1) **Each element is a composite of an index and a position.** In NLP, each word is assigned a unique index from a vocabulary, a conversion handled by a tokenizer; the indices in Figure 1, for instance, are generated by the Llama-3 tokenizer (Grattafiori et al., 2024). A word's position is its sequential location within the sentence (e.g., the word "how" is at position 0 in "how are you"). Similarly, an event's index is its spatial address, represented by the tuple $(x, y, p)$. Crucially, its position is not the sequence number, but its timestamp $t$, which marks its precise temporal location in the event stream.

(2) **The set of possible indices is finite.** The vocabulary of a language, which forms the dictionary used in NLP, is finite. Likewise, an event camera has a limited set of possible event indices, defined by its sensor's properties. For example, a DVS128 camera has $2 \times 128 \times 128$ unique indices, corresponding to 2 polarities across a $128 \times 128$ spatial resolution.

(3) **The sequence exhibits a natural ordering.** Words in a sentence are arranged in a specific sequence that dictates meaning. Analogously, events are naturally ordered by their timestamps, reflecting the chronological progression of captured changes. This inherent temporal order is a key characteristic that distinguishes event data from unordered data structures like point clouds.

(4) **The meaning of an element is determined by its context.** A word can be polysemous; for instance, "transformer" can refer to a neural network architecture or a character in an animated series; its specific meaning is disambiguated by the surrounding text. An individual event merely signals a brightness change at a specific pixel and time, conveying little information in isolation. However, when viewed within a spatiotemporal stream, a sequence of events can delineate an object's contour, thus giving a single event a higher-level meaning, such as being part of an edge. Therefore, the significance of an event is also fundamentally context-dependent.

Inspired by word2vec, we propose event-to-vector (event2vec), an efficient spatio-temporal representation for asynchronous events. Our contributions are as follows:

(1) By embedding events into a vector space, our method natively handles the sparse nature of the input stream, avoiding dense intermediate representations like event frames. This allows for efficient, GPU-accelerated processing with modern network architectures.

(2) We propose a parametric spatial embedding and a convolution-based temporal embedding method that captures neighborhood similarity—a task that is critical for accuracy but difficult for a standard embedding layer to learn.

(3) We validated our method on three classification benchmarks: DVS Gesture, ASL-DVS, and DVS-Lip. It achieved accuracy competitive with traditional methods while demonstrating remarkable parameter efficiency, throughput, and robustness, especially with a low number of events.

Beyond the performance metrics, the most significant contribution of event2vec is its ability to enable neural networks to process event streams in a manner analogous to natural language. Therefore,

state-of-the-art NLP architectures and methods—such as Transformer variants, model acceleration algorithms, and generative self-supervised training—can be directly leveraged for event-based vision. Event2vec paves the way for the native integration of event cameras with large language models and multimodal models.

## 2 RELATED WORK

### 2.1 DENSE REPRESENTATIONS OF EVENTS

Dense representations, derived from raw event streams, are fully compatible with conventional deep learning methods. This is typically achieved by integrating events along the time axis to form dense 3D or 4D tensors, such as event frames (Liu & Delbruck, 2018), multi-channel images (Barchid et al., 2022), voxel grids (Bardow et al., 2016), volumetric cubes (Cordone et al., 2022) and patches (Sabater et al., 2023; Peng et al., 2023).

Specifically, event-to-frame methods accumulate events within discrete time intervals. These resulting frames can then be processed directly by standard neural networks. However, a significant drawback of these methods is the degradation or complete loss of the high temporal resolution inherent to event data. This occurs because individual event timestamps are aggregated or quantized during the conversion process. Furthermore, transforming the data into a dense representation negates the inherent spatial sparsity of events. For instance, the generated frames often contain a substantial number of zero-valued pixels. These pixels, while carrying no information, still incur significant memory and computational overhead. While many methods use timestamps implicitly to define the integration interval, some approaches explicitly leverage them to generate temporal weights (Zhu et al., 2019; Gehrig et al., 2019). Finally, the conversion process itself can be computationally intensive, introducing considerable latency that is often prohibitive for real-time applications (Rebecq et al., 2019; Gallego et al., 2022).

### 2.2 IRREGULAR REPRESENTATIONS OF EVENTS

Conversely, methods for processing irregular representations aim to preserve the inherent sparsity and asynchronicity of event data. This category includes SNNs (Maass, 1997; Roy et al., 2019), Sparse Convolutional Networks (Sparse CNNs) (Messikommer et al., 2020; Santambrogio et al., 2024), Graph Neural Networks (GNNs) (Bi et al., 2019; Schaefer et al., 2022), and point-based methods (Yang et al., 2019; Sekikawa et al., 2019; Lin et al., 2023; Ren et al., 2025).

When deployed on neuromorphic hardware (Merolla et al., 2014; Davies et al., 2018), Spiking Neural Networks (SNNs) can process events in a naturally asynchronous event-driven manner. However, on standard hardware, GPU-based simulations of SNNs produce dense tensor outputs, as the hardware necessitates synchronous processing with discrete time-steps. Consequently, training SNNs on conventional GPUs typically occurs in a synchronous fashion, leading to an unavoidable performance gap between synchronous training and asynchronous inference (Yao et al., 2024; Du et al., 2025). Moreover, the reliance on backpropagation-through-time renders the training process slow and memory-intensive. Sparse CNNs leverage the inherent sparsity of event data, achieving a theoretically low number of Floating-Point Operations (FLOPs). Nevertheless, the architecture of standard GPUs is not optimized for the dynamic computations and unstructured memory access patterns required for efficient sparse acceleration. Consequently, similar to SNNs, Sparse CNNs fail to fully exploit the massive parallel processing capabilities of GPUs.

Event-based GNNs construct graphs from incoming events, an approach that effectively preserves the spatio-temporal relationships between them. Since empty regions with no event activity do not generate graph nodes, the data's sparsity is well-utilized. Their main disadvantage lies in the need for careful hyper-parameter tuning, such as the event downsampling rate and neighborhood radius for graph construction. Additionally, functioning as low-pass filters (Nt & Maehara, 2019), GNNs are susceptible to the over-smoothing problem (Zhou et al., 2020), which limits their ability to form deep architectures comparable to modern CNNs and Transformers (Vaswani et al., 2017). Point-based methods treat events from event cameras as analogous to point clouds from Light Detection and Ranging (LiDAR) sensors. A fundamental limitation of most point cloud models is their permutation invariance, which necessitates treating the input as an unordered set. Consequently, the event timestamp is typically relegated to being an additional positional coordinate, thereby discarding the

crucial causal ordering of events. To manage the data volume, these methods often employ classic point cloud pre-processing techniques like farthest point sampling, which further increases latency.

## 3 METHODS

### 3.1 REPRESENTING EVENTS IN A VECTOR SPACE

Leveraging the strong analogy between words and events, we propose a method for representing events within a vector space, which we term event-to-vector (event2vec). An event, generated by a camera with a spatial resolution of $H \times W$, is represented as a tuple $(x, y, t, p)$. For our embedding, we treat the triplet $(x, y, p)$ as the spatial coordinate and the timestamp $t$ as the temporal coordinate. The general formulation for the event2vec embedding is defined as:

$$\mathbf{v} = \mathbf{v}_s + \mathbf{v}_t = \text{Embed}_s(x, y, p) + \text{Embed}_t(t), \tag{1}$$

where $\mathbf{v} \in \mathbb{R}^D$ is the resulting $D$-dimensional embedding vector, $\mathbf{v}_s = \text{Embed}_s(x, y, p) \in \mathbb{R}^D$ is the spatial embedding module, and $\mathbf{v}_t = \text{Embed}_t(t) \in \mathbb{R}^D$ is the temporal embedding module. As shown in Eq. 1, this method fuses spatial and temporal information through addition. This additive fusion strategy is directly inspired by the positional encoding mechanism prevalent in Transformers.

### 3.2 SPATIAL EMBEDDING

A straightforward approach for the spatial embedding module is to adapt the standard embedding layer from NLP, which is efficiently implemented as a look-up table:

$$\mathbf{v}_s = \text{Embed}_s(x, y, p) = \mathbf{W}_s[p \cdot H \cdot W + y \cdot W + x], \tag{2}$$

where $\mathbf{W}_s \in \mathbb{R}^{(2 \cdot H \cdot W) \times D}$ is the learnable embedding matrix and $D$ is the embedding size. This method maps each unique spatial coordinate to a distinct row index in the embedding matrix $\mathbf{W}_s$.

However, this standard embedding layer imposes no inductive bias on the relationship between indices, compelling the model to learn all spatial relationships from data alone. In a tokenizer, a word's index is a non-semantic identifier, whose order is primarily determined by the token's frequency in the training corpus. Consequently, the words at indices $i$ and $i + 1$ share no inherent semantic similarity. This assumption does not hold for event coordinates. Images are continuous two-dimensional functions (Gonzalez, 2009). Spatially adjacent pixels are known to exhibit strong correlation. Therefore, an effective spatial embedding should incorporate this locality bias, ensuring that events with close coordinates yield similar embedding vectors:

$$\text{Embed}_s(x + \Delta x, y + \Delta y, p) - \text{Embed}_s(x, y, p) \approx \mathbf{0}, \tag{3}$$

for small coordinate perturbations $[\Delta x, \Delta y]$, e.g., $[\Delta x, \Delta y] = [1, 0]$.

The standard embedding in Eq. 2 fails to account for this crucial spatial relationship, which can impede the learning process. To solve this issue, we propose an elegant parametric algorithm for generating the embedding matrix $\mathbf{W}_\phi$ by a neural network $\phi$. To systematically enumerate all spatial coordinates within a $P \times H \times W$ volume (where $P = 2$ represents the two polarities), we first establish a linear index sequence $\mathbf{c} = [0, 1, \ldots, P \cdot H \cdot W - 1]$. This sequence is then decomposed into three probe tensors, $\mathbf{x}_c$, $\mathbf{y}_c$, and $\mathbf{p}_c$, which correspond to the coordinates along the width, height, and polarity dimensions, respectively. The transformation is defined as follows: $\mathbf{x}_c = \mathbf{c} \pmod{W}, \mathbf{y}_c = \left\lfloor \frac{\mathbf{c}}{W} \right\rfloor \pmod{H}, \mathbf{p}_c = \left\lfloor \frac{\mathbf{c}}{WH} \right\rfloor$. Finally, these probe tensors are passed through $\phi$, which outputs the complete embedding matrix $\mathbf{W}_\phi = \phi(\mathbf{x}_c, \mathbf{y}_c, \mathbf{p}_c)$. By substituting the parametrically generated matrix $\mathbf{W}_\phi$ into the look-up mechanism of Eq. 2, we establish a direct equivalence for any given event coordinate $(x, y, p)$:

$$\mathbf{W}_\phi[p \cdot H \cdot W + y \cdot W + x] = \phi(x, y, p). \tag{4}$$

Crucially, the parametric network $\phi$ is designed to be a continuous and differentiable function. This property allows us to formally analyze the relationship between neighboring embeddings using a first-order Taylor series expansion:

$$\phi(x + \Delta x, y + \Delta y, p) - \phi(x, y, p) \approx \frac{\partial \phi}{\partial x} \Delta x + \frac{\partial \phi}{\partial y} \Delta y + o(\|\Delta\|), \tag{5}$$

where $o(\|\Delta\|)$ represents higher-order remainder terms. As Eq. 5 illustrates, for small perturbations $[\Delta x, \Delta y]$, the difference between the embeddings is proportional to the gradient of $\phi$. Consequently, as the perturbations approach zero, this difference vector also approaches zero:

$$\phi(x + \Delta x, y + \Delta y, p) - \phi(x, y, p) \approx \mathbf{0}. \tag{6}$$

In this manner, the use of a continuous parametric network $\phi$ inherently embeds the desired neighborhood semantics, or spatial inductive bias, directly into the embedding matrix. This approach elegantly satisfies the condition outlined in Eq. 3.

### 3.3 TEMPORAL EMBEDDING

Timestamps, which denote the occurrence time of events, serve a function analogous to positional indices in a sentence. In modern NLP models, relative positional encoding methods (Press et al., 2021; Su et al., 2024) are increasingly favored over absolute methods, such as sinusoidal encoding (Vaswani et al., 2017) or learnable absolute positional embeddings (Devlin et al., 2019b).

However, directly applying these relative positional encoding techniques to event timestamps is ill-suited. Such methods are fundamentally designed for discrete and uniformly spaced indices, whereas event timestamps are continuous and inherently non-uniform. To address this discrepancy, we propose learning the temporal embedding directly from the differences between consecutive timestamps using a neural network.

Specifically, the temporal embedding module is implemented as a stack of convolutional layers. For the $i$-th event in an event stream, the input to this module is the first-order temporal difference, $t[i+1] - t[i]$. This design offers several distinct advantages:

(1) **Time-Shift Invariance:** By operating on relative temporal distances, the embedding becomes inherently invariant to absolute shifts in time.

(2) **Generalization:** It functions as a variant of position-wise learnable encoding but circumvents the length generalization problem by accepting continuous values as input, rather than being tied to a fixed vocabulary of discrete positions.

(3) **Contextual Consistency:** The convolutional operations allow the temporal embedding for an event to be influenced by the timing of its immediate neighbors, thereby reinforcing the principle of neighborhood semantics in the time domain.

### 3.4 EVENT SAMPLING AND AGGREGATION

Raw event streams often contain an extremely large number of events, with sequence lengths exhibiting substantial variance. Furthermore, deep learning frameworks typically process data in batches, which requires that all tensors within a single batch have uniform dimensions. Consequently, it is necessary to sample or aggregate events from each stream to a fixed-length sequence of size $L$.

In this paper, we primarily use two methods. The first is uniform random sampling. We find that this straightforward method works well in most cases and is extremely computationally efficient. However, a significant limitation of random sampling is the substantial information loss incurred by discarding the majority of the events, leading to suboptimal accuracy in complex tasks. Our second method addresses this by leveraging k-means clustering to aggregate the entire event stream into $L$ representative clusters. Specifically, the clustering process is performed independently on the two event polarities to preserve their distinct information channels. After clustering, the centroid of each cluster is treated as a representative event. Its timestamp is taken directly from the centroid's temporal coordinate, while its spatial coordinates are quantized to the nearest integers. Furthermore, we compute an intensity factor, $\rho$, equal to the number of raw events belonging to that cluster. This intensity factor then modulates the corresponding spatial embedding vector, effectively weighting the representation by its event density. This approach ensures that information from every event contributes to the final representation.

In summary, the final event2vec representation for a sequence of $L$ events is a tensor $\mathbf{V} \in \mathbb{R}^{L \times D}$. The embedding for the $i$-th event in this sequence, $\mathbf{V}[i]$, is formulated as:

$$\mathbf{V}[i] = \boldsymbol{\rho}[i] \cdot \text{Embed}_s(\mathbf{x}[i], \mathbf{y}[i], \mathbf{p}[i]) + \text{Embed}_t(\Delta \mathbf{t})[i], \tag{7}$$

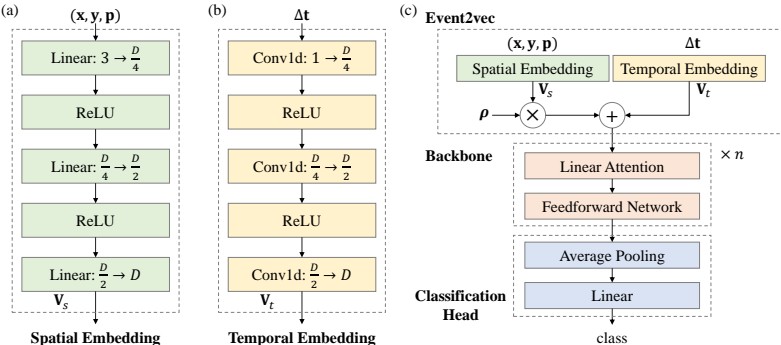

Figure 2: The network architecture for event classification using the event2vec representation.

where $\boldsymbol{\rho}$, $\mathbf{x}$, $\mathbf{y}$, $\mathbf{p}$, and $\mathbf{t}$ are vectors representing the intensity factors, spatial coordinates, and timestamps for the sequence of $L$ events. The vector of temporal differences is defined as $\Delta\mathbf{t}[i] = \mathbf{t}[i+1] - \mathbf{t}[i]$, with the final value set to zero. For a native event, $\boldsymbol{\rho}[i]$ is 1, while for a cluster event, it represents the number of raw events aggregated into that cluster.

### 3.5 NETWORK STRUCTURE

In this paper, we employ the Transformer architecture, leveraging its core strengths: the ability to efficiently process sequences in parallel and capture long-range dependencies. These characteristics make it particularly well-suited for the sequential representations generated by event2vec.

As shown in Figure 2(a), the spatial embedding module $\phi$ consists of a stack of linear layers. It gradually increases the number of features from 3 to $D$. Layer Normalization (Ba et al., 2016) layers are also inserted to stabilize training but omitted from the figure for clarity. The temporal embedding module has a similar structure to the spatial embedding module, except that it uses depth-wise convolutional layers with a kernel size of 3 and a stride of 1, shown in Figure 2(b).

We employ the Forgetting Transformer (Lin et al., 2025) as the linear attention in Figure 2(c). It is important to recognize that linear attention is fundamentally equivalent to RNNs (Katharopoulos et al., 2020), operating with fixed-size hidden states. To enhance the learning capability for extremely long event sequences, we extend the linear attention model to a parameter-shared bidirectional formulation. Further details are provided in Appendix A.2. A linear attention module and a feedforward network, consisting of two linear layers, together constitute a single backbone block. As illustrated in Figure 2(c), the full backbone is composed of $n$ such blocks stacked sequentially. For the classification head, we employ an average pooling layer to aggregate features across all positions in the sequence.

## 4 EXPERIMENTS

To validate the event2vec representation, we conduct a series of experiments on classification tasks using three neuromorphic datasets: DVS Gesture (Amir et al., 2017), ASL-DVS (Bi et al., 2019), and DVS-Lip (Tan et al., 2022). In this section, results are reported in the format $a \pm b$, representing the mean and standard deviation, respectively. For experiments that involve random sampling, these statistics are computed over 10 independent runs on the test set. Experimental details are provided in Appendices A.3 to A.8.

### 4.1 COMPARISON BETWEEN REPRESENTATIONS

**Accuracy and Parameter Efficiency** Table 1 compares the accuracy and model parameters of event2vec with those of other representations across the three datasets. Our models for DVS Gesture and ASL-DVS are trained directly on randomly sampled events. For DVS-Lip, our model first undergoes self-supervised pre-training (refer to Appendix A.5) on cluster events. We then report the fine-tuning accuracy on both randomly sampled events and cluster events. Our method achieves accuracy comparable to that of other leading representations while demonstrating exceptional parameter efficiency.

Table 1: Model performance and size comparison on neuromorphic datasets

| Dataset | Method + Representation | Accuracy (%) | Params (MB) |
|---|---|---|---|
| DVS Gesture | Sparse GRU + Frame (Subramoney et al., 2023) | 97.80 | 4.8 |
| | SNN + Frame (Yao et al., 2023) | **98.23** | 6.5 |
| | FARSE-CNN + Window Slicing (Santambrogio et al., 2024) | 96.6 | 10.79 |
| | Event MAE + Point Cloud (Sun et al., 2025) | 97.75 | Unknown |
| | Linear Attention + Event2vec (4096 Randomly Sampled Events) | 97.57±1.31 | **0.52** |
| ASL-DVS | GNN,CNN + Graph (Bi et al., 2019) | 90.10 | 19.46 |
| | GNN,Transformer + Image,Voxel Graph (Yuan et al., 2023) | 99.60 | 220.3 |
| | Linear Attention + Event2vec (512 Randomly Sampled Events) | **99.91±0.05** | **0.27** |
| DVS-Lip | ResNet-18,BiGRU + Frame (Tan et al., 2022) | 60.3 | 72.1 |
| | Spiking ResNet18,BiGRU + Frame (Dampfhoffer & Mesquida, 2024) | **75.3** | 58.6 |
| | Linear Attention + Event2vec (1024 Randomly Sampled Events) | 70.62±1.55 | **18.3** |
| | Linear Attention + Event2vec (1024 Cluster Events) | 75.14 | **18.3** |

Table 2: Comparative analysis of pre-processing latency for event representations on DVS Gesture

| Representation | Hyper-Parameter | Total Pre-processing Time (s) |
|---|---|---|
| Random Sampling (used in event2vec) | 4096 events | **1.24±0.04** |
| Frame (Yao et al., 2023) | 16 frames | 4.44±0.03 |
| Graph (Schaefer et al., 2022) | Radius=5, 32 neighbors, 10000 samples | 6.37±0.06 |
| Voxel Grid (Zhu et al., 2019) | 16 bins | 7.70±0.15 |
| Point Cloud (Sun et al., 2025) | 64 patches × 32 points | 47.56±1.50 |
| K-Means (used in event2vec) | 1024 clusters | 119.51±0.55 |
| Window Slicing (Santambrogio et al., 2024) | Size=100ms, stride=20ms | 230.79±2.46 |

Table 3: Benchmark of throughput and single event stream inference latency on DVS Gesture

| Method | Throughput (samples/s) Training | Inference | Single Stream Latency (ms) |
|---|---|---|---|
| FARSE-CNN + Window Slicing (Santambrogio et al., 2024) | 6.65±0.85 | 23.94±9.21 | 324.16±111.15 |
| AEGNN + Graph (Schaefer et al., 2022) | 111.06±6.68 | 1271.75±2.69 | 3.74±0.20 |
| SNN + Frame (Yao et al., 2023) | 202.88±3.69 | 234.00±7.61 | 6.12±0.34 |
| Sparse GRU + Frame (Subramoney et al., 2023) | 417.00±0.72 | 472.82±18.40 | **2.71±0.02** |
| Event2vec + Randomly Sampling | **1030.02±56.36** | **2884.56±283.89** | 11.44±0.69 |

**Event Pre-processing Time** Table 2 benchmarks the total pre-processing time required by different representations to process all 1,176 event streams of the DVS Gesture training set. Hyper-parameters were set to the values specified in their respective original publications, where available. For others, we adopted commonly used values; for instance, graph construction was constrained to an average node degree between 10 and 20 to ensure balanced connectivity, and the number of frames and bins of the voxel grid was set to 16, a common configuration (Zhou et al., 2024). The results indicate that event2vec with random sampling exhibits minimal pre-processing time. While employing K-Means clustering increases pre-processing time, it still remains faster than window slicing.

**Throughput and Latency** Table 3 compares the throughput and single event stream inference latency of FARSE-CNN, AEGNN, SNN, Sparse GRU and our event2vec model on the DVS Gesture dataset. Our event2vec model exhibits extremely high training and inference throughput, primarily due to its full compatibility with a highly optimized linear attention library that can fully leverage GPUs for acceleration. The FARSE-CNN model exhibits the lowest throughput. This is attributable to its reliance on Sparse CNN, which receives limited acceleration from unstructured sparsity on GPUs, and the inclusion of a recurrent structure that further constrains its speed.

## 4.2 ABLATION EXPERIMENTS

**Embedding Comparison** We conducted an ablation study on the DVS Gesture dataset to evaluate the accuracy contributions of different components, as detailed in Table 4. We tested various combinations of spatial embedding methods (standard (Eq. 2) vs. parametric (Eq. 4)) and temporal embedding modules (sinusoidal embedding on $t$ vs. convolutional embedding on $\Delta t$). The combination of the standard embedding with our convolutional temporal

Table 4: Ablation analysis of embeddings on the DVS Gesture dataset

| Embedding Spatial | Temporal | Accuracy (%) |
|---|---|---|
| Standard | Conv($\Delta t$) | 91.18±3.70 |
| Standard | Sin($t$) | 93.16±2.19 |
| Parametric | Sin($t$) | 96.56±1.46 |
| Parametric | Conv($\Delta t$) | 97.57±1.31 |

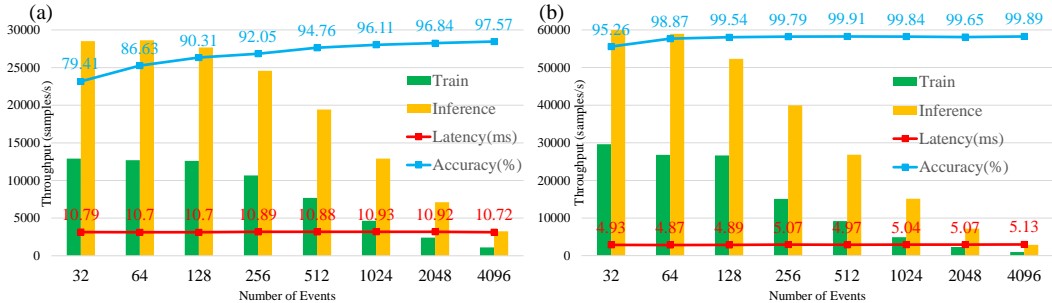

Figure 3: Effect of number of events on models for (a) DVS Gesture (b) ASL-DVS.

embedding (Standard + Conv($\Delta t$)) yields the lowest accuracy. We attribute this to the standard embedding layer's lack of inductive bias, which prevents it from effectively learning neighborhood semantics and subsequently hinders the convolutional temporal encoder. Consequently, when using our parametric embedding, the convolutional encoder achieves the highest accuracy. It is worth noting that our parametric embedding consistently outperforms the standard version when paired with any temporal embedding, validating the effectiveness of incorporating neighborhood semantics.

**Effect of Event Numbers** Processing fewer events results in lower resource consumption, which is always desirable in event-based applications. We benchmark the impact of varying the number of randomly sampled events ($L$) on several key metrics: training/inference throughput, single event stream inference latency, and accuracy on the DVS Gesture and ASL-DVS datasets. As illustrated in Figure 3, the performance trends are consistent across both datasets. For small values of $L$, throughput decreases only marginally as $L$ increases. This is attributable to the CUDA kernel launch overhead, which dominates the computation time, rendering the actual kernel execution time negligible in this regime. As $L$ grows larger, the kernel execution time becomes the primary bottleneck, causing a more pronounced decrease in throughput. Notably, while $L$ increases, the throughput decreases approximately inversely proportionally, a finding consistent with the $\mathcal{O}(L)$ complexity of linear attention. The single event stream inference latency for both models increases only slightly with $L$, further indicating that kernel launch overhead, rather than execution time, remains the dominant factor. In terms of accuracy, the general trend shows improvement as $L$ increases. Surprisingly, both models maintain a reasonable level of accuracy even with a very small number of events (e.g., $L = 32$), demonstrating their robustness to sparse inputs.

We also compare our method with the sophisticated sampling techniques from Araghi et al. (2025), which use a voxel grid representation (Figure 4). The results highlight the inherent effectiveness of event2vec: when paired with simple random sampling, it consistently outperforms the voxel grid representation, even when the latter employs more complex, meticulously designed sampling strategies.

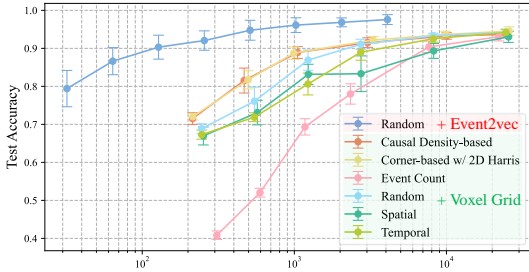

Figure 4: DVS Gesture: Accuracy vs. number of events against Araghi et al. (2025).

### 4.3 VISUALIZATION

**Neighborhood Semantics** To visually inspect the neighborhood semantics, we extract the spatial embedding weights from models trained on the DVS Gesture dataset with the parametric ($\mathbf{W}_\phi$) and standard ($\mathbf{W}_s$) embedding layers. For each coordinate $(x, y, p)$, its $D$-dimensional embedding vector is projected onto a 3-dimensional space using Principal Component Analysis (PCA). These 3D vectors are then interpreted as RGB color values and plotted at their corresponding $(x, y)$ locations to form an image. Figure 5(a) visualizes the resulting images for polarity 0 (images for polarity 1 are provided in Appendix A.9). The image derived from $\mathbf{W}_\phi$ displays smooth, continuous color gradients, akin to a color palette, indicating that spatially adjacent coordinates have semantically similar embeddings. In stark contrast, the image from $\mathbf{W}_s$ resembles random noise, signifying a lack of learned spatial correlation.

**Polarity Similarity** An object's edge moving across a pixel often triggers events of both polarities in close succession. We therefore hypothesize that the embeddings for opposite polarities at the same

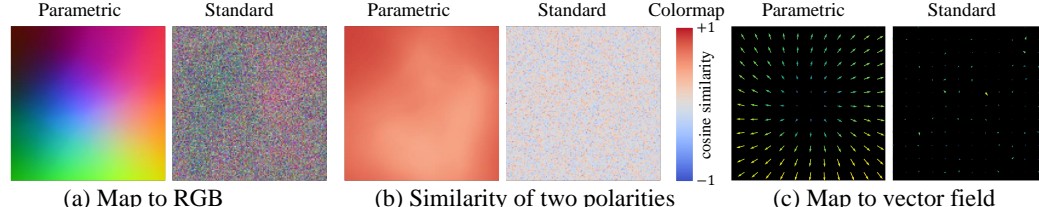

| Parametric | Standard | Parametric | Standard | Colormap | Parametric | Standard |

(a) Map to RGB       (b) Similarity of two polarities       (c) Map to vector field

Figure 5: Visual comparison of the learned spatial embeddings: parametric vs. standard.

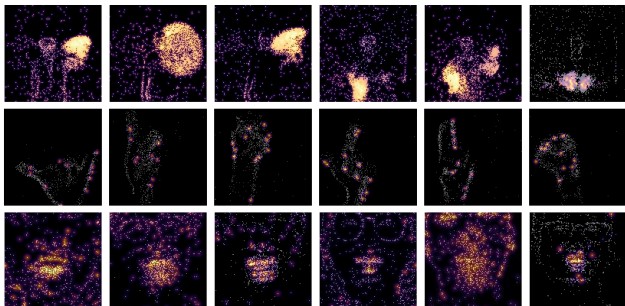

Figure 6: Event-level attention maps on samples from DVS Gesture (Row 1), ASL-DVS (Row 2), and DVS-Lip (Row 3).

spatial location should also be semantically related. To test this, we compute the cosine similarity between the embedding vectors of the two polarities at each coordinate. As shown in Figure 5(b), the parametric embedding captures this relationship, exhibiting distinct regions of high similarity. Conversely, the similarity map for the standard embedding is predominantly close to zero, indicating that it fails to learn this inter-polarity correlation.

**Vector Field Representation** We visualize the learned spatial manifold as a vector field. The $D$-dimensional embedding vectors are projected onto their first two principal components using PCA. These resulting 2D vectors are then visualized using a quiver plot, where each arrow represents the direction and magnitude of the vector at its spatial coordinate. Figure 5(c) illustrates the results. The vector field for the parametric embedding exhibits a coherent, laminar-like flow, revealing a smoothly structured semantic space. In contrast, the field for the standard embedding appears chaotic and turbulent, further confirming its inability to capture meaningful spatial relationships.

**Event-wise Attention** As event2vec is an event-wise representation, its attention mechanism can be visualized at a fine-grained, event-level resolution. Figure 6 displays attention heatmaps overlaid on the original event streams for DVS Gesture (row 1), ASL-DVS (row 2), and DVS-Lip (row 3). The visualizations reveal that the model correctly focuses on the hands in DVS Gesture, the finger joints and contours in ASL-DVS, and the lip region in DVS-Lip. However, consistent with the lower classification accuracy compared to the other two datasets, we also observe instances where the model incorrectly allocates significant attention to other facial features, such as the eyes and ears.

## 5 CONCLUSIONS

Neuromorphic event cameras introduce a paradigm shift in computer vision, presenting both unique opportunities and significant challenges. A central challenge has been reconciling their asynchronous, sparse nature with the synchronous, dense tensor-based architectures of deep learning. In this paper, we introduced event2vec, a novel representation that directly addresses this challenge by enabling neural networks to natively process asynchronous events. Our experimental results demonstrate that event2vec achieves accuracy competitive with established methods while offering compelling advantages in parameter efficiency, pre-processing overhead, throughput, and robustness across varying numbers of events. Beyond these performance metrics, the most significant contribution of event2vec is its conceptual alignment of event streams with the paradigm of natural language processing. This opens new avenues for research and application. By treating events as a sequential language, we can begin to explore novel applications by leveraging the sophisticated architectures developed for large language models.

## REPRODUCIBILITY STATEMENT

The experimental code, training logs, terminal output, and trained models for this paper are all provided together in the code repository `https://anonymous.4open.science/r/event2vec_iclr-7B40`. We have included detailed instructions in the repository, allowing readers to easily prepare the dataset and reproduce the experiments based on these instructions. All experiments in this paper fix the random number seeds of PyTorch, NumPy, and Python to 0 using the `seed_everything` function from PyTorch Lightning, in order to maintain reproducibility as much as possible.

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

## A  APPENDIX

### A.1  DATASETS

The DVS Gesture dataset is a benchmark commonly employed for model evaluation. It comprises 11 categories of hand gestures and is officially divided into a training set with 1,176 samples and a testing set with 288 samples.

The ASL-DVS dataset contains 24 classes corresponding to letters from American Sign Language, amounting to a total of 100,800 samples. Each class consists of 4,200 samples, and the duration of each sample is approximately 100 ms. Following the methodology in Bi et al. (2019), we partition the dataset by allocating the initial 80% for training and the remaining 20% for testing.

The DVS-Lip dataset encompasses 100 word classes derived from the visual information of a speaker's lip movements. It provides an official training/testing split with 14,896 and 4,975 samples, respectively.

Table 5: Hyper-parameters of training models for classification tasks on different datasets

| Dataset | $D$ | $D_f$ | $n_{head}$ | Depth | Repeats | $n_{gpus}$ | $lr_{min}$ |
|---|---|---|---|---|---|---|---|
| DVS Gesture | 64 | 128 | 2 | 4 | 24 | 4 | 0 |
| ASL-DVS | 64 | 128 | 2 | 2 | 1 | 7 | $10^{-6}$ |
| DVS-Lip | 192 | 384 | 6 | 16 | 3 | 4 | $10^{-6}$ |

## A.2 BI-DIRECTIONAL LINEAR ATTENTIONS

Ignoring the layer index for simplicity, the recurrence relation for the hidden state $\mathbf{S}[t]$ at time-step $t$ in a standard linear attention model is:

$$\mathbf{S}[t] = f(\mathbf{S}[t-1], \mathbf{k}[t], \mathbf{v}[t]), \tag{8}$$
$$\mathbf{O}[t] = \mathbf{W}_o \mathbf{S}[t], \tag{9}$$

where $\mathbf{k}[t]$ and $\mathbf{v}[t]$ are input-dependent key and value vectors, $\mathbf{W}_o$ is the output projection matrix, and $\mathbf{O}[t]$ is the output vector. The specific recurrence function, $f$, used in this work is based on the Forgetting Transformer (FOX) (Lin et al., 2025) implemented by the Flash Linear Attention library (Yang & Zhang, 2024)

We adapt this formulation to be bi-directional by maintaining separate forward and backward hidden states, which are then concatenated and fused through a linear layer to produce the output $\mathbf{O}_{fb}[t]$:

$$\mathbf{S}_f[t] = f(\mathbf{S}_f[t-1], \mathbf{k}[t], \mathbf{v}[t]), \tag{10}$$
$$\mathbf{S}_b[t] = f(\mathbf{S}_b[t-1], \mathbf{k}[L-t-1], \mathbf{v}[L-t-1]), \tag{11}$$
$$\mathbf{S}_{fb}[t] = [\mathbf{S}_f[t]; \mathbf{S}_b[t]], \tag{12}$$
$$\mathbf{O}_{fb}[t] = \mathbf{W}_{fb} \mathbf{S}_{fb}[t]. \tag{13}$$

Unlike classic bi-directional RNNs (Schuster & Paliwal, 1997) that often use independent parameters for each direction, our model employs shared parameters for the forward and backward passes. Consequently, the only increase in parameters compared to the uni-directional model arises from the output projection matrix $\mathbf{W}_{fb}$, which has twice the number of parameters as the original $\mathbf{W}_o$.

## A.3 MODEL HYPER-PARAMETERS

Unless otherwise stated, all models were trained using BFloat16 mixed precision. The training configuration for all models includes a base learning rate of $lr_b = 0.001$, a batch size of 64, and the AdamW optimizer (Loshchilov & Hutter, 2019), conducted over 64 epochs. The effective learning rate is determined by a linear scaling rule based on the number of GPUs ($n_{gpus}$) used in distributed data-parallel training: $lr = lr_b \cdot n_{gpus}/256$. A warmup phase is implemented for the first four epochs, during which the learning rate is linearly increased from $0.01 \cdot lr$ to its full value, $lr$. For the subsequent epochs, a cosine annealing schedule (Loshchilov & Hutter, 2017) is employed to gradually reduce the learning rate to a minimum value, $lr_{min}$. For the DVS Gesture and ASL-DVS datasets, both weight decay and label smoothing were disabled. In contrast, for the DVS-Lip classification task, we set the weight decay to 0.05 and applied label smoothing with a factor of 0.1.

Table 5 provides a detailed summary of the model-specific hyper-parameters. Here, $D$ denotes the embedding dimension, $D_f$ represents the hidden feature dimension of the feed-forward neural network (FFN), and $n_{head}$ is the total number of attention heads. The `repeats` parameter specifies how many times the training set is iterated through within a single epoch. Notably, the number of heads for the key ($\mathbf{k}$) and value ($\mathbf{v}$) projections is set to $n_{head}/2$, and group normalization (Wu & He, 2018) is applied to both. To prevent exploding gradients, we employ gradient clipping, capping the $L_2$ norm of the gradients at 1.0.

For the DVS Gesture classification model, the output of each FFN is average-pooled with a window size of 2, whereas other models do not use pooling. The model for the DVS-Lip classification task was pre-trained on the DVS-Lip dataset using a self-supervised learning approach. This pre-training phase utilized a minimum learning rate of $lr_{min} = 10^{-6}$, a weight decay of 0.05, a `repeats` value of 3, and a masking ratio of 30%. Refer to Appendix A.5 for more details.

## A.4 DATA AUGMENTATIONS

Denote $\mathcal{U}(a, b)$ as the uniform distribution between $a$ and $b$, and $\text{RandInt}(m, n)$ as a random integer taken from the set $m, m + 1, ..., n$, where each integer has an equal probability of being selected.

For an event stream, the data augmentations are applied on events directly. For simplicity, we omit the event index in this subsection. Unless otherwise specified, augmentations are applied on each event stream independently. Note that the coordinates are converted to floating percision before applying any augmentation. After all augmentations are applied, coordinates will be quantized, and only events whose coordinates are valid, i.e., $x \in [0, W - 1], y \in [0, H - 1]$, are kept.

For the classification task on DVS Gesture, the following transformations are each applied independently with a probability of 0.6:

- **Random Resizing**: Coordinates $(x, y)$ are scaled to $(s_x \cdot x, s_y \cdot y)$, with scaling factors $s_x, s_y \sim \mathcal{U}(0.8, 1.2)$.
- **Random Rotation**: Coordinates are rotated by an angle $r \sim \mathcal{U}(-10, 10)$ degrees.
- **Random Shearing**: A shear transformation is applied with factors $\lambda_x, \lambda_y \sim \mathcal{U}(-0.02, 0.02)$.
- **Random Translation**: Coordinates are translated by offsets $d_x, d_y \sim \mathcal{U}(-16, 16)$.
- **Random Erasing**: Erase an $h \times w$ area with $h, w \sim \mathcal{U}(0, 16)$ with the probability 0.1. The center of this area $(c_x, c_y)$ satisfy $c_x \sim \mathcal{U}(0, W - 1), c_y \sim \mathcal{U}(0, H - 1)$.
- **Temporal Chunk Dropout**: A number of temporal chunks, $n_r = \text{RandInt}(0, 8)$, are removed from the event stream. The length of each removed chunk, $l_{chunk}$, is determined relative to the total stream length, $L$, according to the sampling distribution $l_{chunk} = \frac{\text{RandInt}(1,256)}{L}$.

No data augmentations were applied for the ASL-DVS dataset.

During the self-supervised phase of the model for classifying DVS-Lip, a series of geometric transformations are employed. Each of the following augmentations is applied independently with a probability of 0.5:

- **Random Resizing**: Coordinates $(x, y)$ are scaled to $(s_x \cdot x, s_y \cdot y)$, with scaling factors $s_x, s_y \sim \mathcal{U}(0.8, 1.2)$.
- **Random Rotation**: Coordinates are rotated by an angle $r \sim \mathcal{U}(-15, 15)$ degrees.
- **Random Shearing**: A shear transformation is applied with factors $\lambda_x, \lambda_y \sim \mathcal{U}(-0.05, 0.05)$.
- **Horizontal Flipping**: The event stream is flipped horizontally.
- **Random Translation**: Coordinates are translated by offsets $d_x, d_y \sim \mathcal{U}(-16, 16)$.

When training the model for classifying DVS-Lip, we use the following data augmentations:

- **Random Resizing**: Resize $(x, y)$ to $(s_x \cdot x, s_y \cdot y)$ where $s_x, s_y \sim \mathcal{U}(0.8, 1.2)$.
- **Random Rotation**: Coordinates are rotated by an angle $r \sim \mathcal{U}(-15, 15)$ degrees.
- **Random Shearing**: Shear transform on $x$ and $y$ with shear factors $\lambda_x, \lambda_y \sim \mathcal{U}(-0.05, 0.05)$.
- **Random Flip**: The event stream is flipped horizontally with a probability of 0.5.
- **Random Translation**: Translate $x$ and $y$ with translations $d_x, d_y \sim \mathcal{U}(-16, 16)$.
- **Random Erasing**: Erase an $h \times w$ area with $h, w \sim \mathcal{U}(0, 16)$ with the probability 0.1. The center of this area $(c_x, c_y)$ satisfies $c_x \sim \mathcal{U}(0, W - 1), c_y \sim \mathcal{U}(0, H - 1)$.

The augmentations listed above are each applied independently with a probability of 0.5. The token-mix is applied on the embedding tensor with probability 0.5. Specifically, when training on cluster events, the intensity $\rho$ is randomly set to 1 with a probability of 0.1. We use drop path (Larsson et al., 2016) in the linear attention layer, with the probability increasing linearly from 0 to 0.4 with depth.

### A.5 SELF-SUPERVISED TRAINING DETAILS

The event-wise nature of the event2vec representation lends itself well to self-supervised pre-training, which can significantly enhance model performance. Specifically, we adopt a masked modeling approach, akin to that used in BERT. The training objective is to mask the spatial coordinates $(x, y, p)$ of a subset of these events and train the model to predict the masked coordinates based on the context provided by the surrounding events and their associated temporal information. This task compels the model to learn a meaningful understanding of spatio-temporal event patterns.

The self-supervised training framework is analogous to the Masked Language Model (MLM) objective in BERT (Devlin et al., 2019a). Given a batch of embedding tensors $\mathbf{v}$ of shape $(B, L, D)$, where $B$ is the batch size, $L$ is the sequence length, and $D$ is the embedding dimension, the process begins by randomly masking a portion of the input tokens.

A binary mask $\mathbf{m}$ of shape $(B, L)$ is generated from a Bernoulli distribution. The probability of masking any given token is set to 0.3, which defines the mask ratio. Each token $\mathbf{v}[i][j]$ corresponding to a mask entry $\mathbf{m}[i][j] = 1$ is replaced by a single, learnable, $D$-dimensional mask token $\mathbf{v}_m$. This operation results in a corrupted embedding tensor, denoted as $\hat{\mathbf{v}}$. Concurrently, the original coordinates $(\mathbf{x}_m, \mathbf{y}_m, \mathbf{p}_m)$ of the masked tokens are preserved to serve as the ground truth for the reconstruction loss.

The corrupted tensor $\hat{\mathbf{v}}$ is then processed by the model's linear attention layers. Following this, the output embeddings that correspond to the initially masked positions, denoted $\hat{\mathbf{v}}_m$, are extracted from the final output tensor using the mask $\mathbf{m}$.

The objective is for the model to reconstruct the original spatial and polarity information from these corrupted embeddings. To achieve this, we first apply the inverse of the spatio-temporal fusion operation to isolate the spatial component of the reconstructed embeddings:

$$\hat{\mathbf{v}}_s = \frac{\hat{\mathbf{v}}_m}{\boldsymbol{\rho}} - \mathbf{v}_t. \tag{14}$$

The resulting tensor, $\hat{\mathbf{v}}_s$, is treated as the reconstructed spatial embedding. It is then passed through a decoder network, which mirrors the architecture of the spatial embedding encoder, to predict the original coordinates $(\hat{\mathbf{x}}, \hat{\mathbf{y}}, \hat{\mathbf{p}})$. Specifically, this decoder consists of a stack of linear layers, Layer Normalization, and ReLU activation functions. The network is designed to gradually reduce the feature dimension from $D$ down to 3. The final output layer uses a tanh activation function to constrain the predicted values to the range $(-1, 1)$. This aligns with the input preprocessing, where the ground-truth coordinates are also normalized to the same range.

Finally, the training objective is to minimize the Mean Squared Error (MSE) loss between the predicted coordinates $(\hat{\mathbf{x}}, \hat{\mathbf{y}}, \hat{\mathbf{p}})$ and the ground-truth coordinates $(\mathbf{x}_m, \mathbf{y}_m, \mathbf{p}_m)$ of the masked tokens.

### A.6 EXPERIMENTAL DETAILS FOR PRE-PROCESSING LATENCY

The pre-processing latency benchmarks, with results presented in Table 2, were conducted on a Red Hat Enterprise Linux 8.10 server. This server was equipped with an NVIDIA A100 GPU (80GB PCIe), an Intel Xeon Gold 6326 CPU (utilizing 8 cores), and 256GB of RAM. To mitigate the impact of data I/O, the DVS Gesture dataset was loaded entirely into RAM for the duration of the experiments. For each method, we optimized the batch size and the number of workers to achieve the minimum possible latency. All implementations were based on the publicly available source code from their respective original publications or other high-performance software libraries. The reported latency for each method is the average of 8 measurement runs, which were preceded by 2 warm-up runs to ensure system stability. Further implementation details are provided in Table 6.

### A.7 EXPERIMENTAL DETAILS FOR MODEL THROUGHPUT AND LATENCY

The throughput and latency experiments, with results presented in Table 3, were conducted under the same operating system and hardware environment as the pre-processing benchmarks, which are detailed in Appendix A.6. The dataset was also pre-loaded into RAM to eliminate I/O bottlenecks.

For each model, we performed a search for the optimal batch size to maximize performance. The optimal batch sizes were determined to be 64 for Sparse GRU, 16 for FARSE-CNN, 64 for AEGNN,

Table 6: Implementation details of Table 2

| Method | Batch size | Workers | Software Library |
|---|---|---|---|
| Random Sampling | 64 | 64 | Numpy (Harris et al., 2020b) |
| Frame | 8 | 8 | SpikingJelly (Fang et al., 2023) |
| Graph | 8 | 8 | PyTorch Geometric (Fey & Lenssen, 2019) |
| Voxel Grid | 16 | 8 | Tonic (Lenz et al., 2021) |
| K-Means | 2 | 2 | Faiss (Johnson et al., 2019) |
| Window Slicing | 8 | 8 | Original paper (Santambrogio et al., 2024) |

64 for SNN and 512 for event2vec, respectively. We observed that a fixed number of 8 workers yielded the best performance across all models. The benchmarking process for each model involved an initial warm-up phase, followed by multiple measurement runs, the results of which were then averaged. Due to significant variations in computational cost among the models, the number of warm-up iterations and measurement batches was tailored for each specific model. However, we ensured that the number of batches was sufficiently large, such that further increases did not yield any significant changes in the measured performance, confirming the stability of our results.

Regarding the implementations, for Sparse GRU[1] and FARSE-CNN[2], the complete source code was available in the official GitHub repositories, and we used them directly. For AEGNN, only the inference code was publicly available[3]. We therefore implemented the necessary `Dataset` class and model architecture by referencing their provided implementation for the N-Cars dataset (Sironi et al., 2018). For SNNs, the model used by Yao et al. (2023) enhances the Parametric Leaky Integrate-and-Fire Neuron Network (PLIF-Net) (Fang et al., 2021) with attentions, but the source code is not released. Given these additional attention modules only add slight complexity, we evaluate on the PLIF-Net as an alternative. SpikingJelly (Fang et al., 2023) provides a high-performance implementation[4] for the PLIF-Net with advanced accelerating techniques, and we benchmark on code from SpikingJelly directly.

All benchmarks were running in BFloat16 mixed precision except for Sparse GRU, which depends on the Haste library (Nanavati, 2020) with only supports Float32.

## A.8 EXPERIMENTAL DETAILS FOR THE ABLATION STUDY ON EVENT NUMBERS

The ablation study on the number of events, with results reported in Figure 3, evaluates the impact on model throughput, latency, and accuracy. These experiments were conducted using the same operating system and hardware environment detailed in Appendix A.6. During the training process on the DVS Gesture dataset, the chunk length parameter $l_{chunk}$ for the temporal chunk dropout augmentation is scaled proportionally with the number of events. Furthermore, for experiments using 64 and 32 events, all data augmentation techniques are disabled. This measure is implemented to prevent the augmentation from inadvertently removing all events, which would subsequently lead to a NaN (Not-a-Number) loss.

## A.9 VISUALIZATION OF NEIGHBORHOOD SEMANTICS

Due to space constraints in the main paper, Figures 5(a) and 5(c) display visualizations for only a single event polarity. For completeness, this section provides supplementary visualizations that include both polarities. Figure 7 illustrates the embedding weights mapped to the RGB color space, while Figure 8 depicts them as a vector field.

---

[1]https://github.com/Efficient-Scalable-Machine-Learning/EvNN
[2]https://github.com/AIRLab-POLIMI/farse-cnn
[3]https://github.com/uzh-rpg/aegnn
[4]https://spikingjelly.readthedocs.io/zh-cn/latest/activation_based_en/classify_dvsg.html

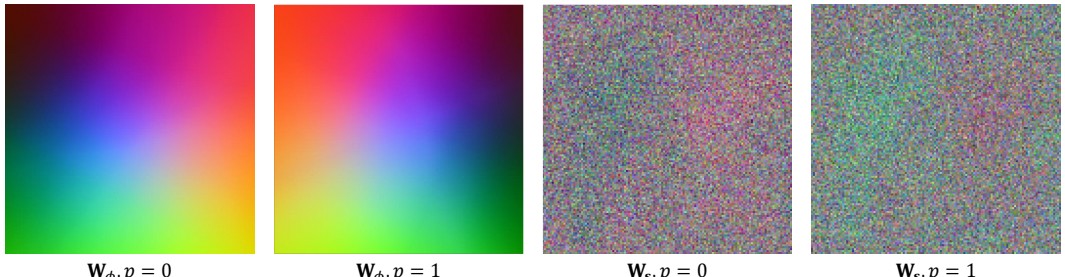

Figure 7: Visualization of the parametric embedding weight $\mathbf{W}_\phi$ and the standard embedding weight $\mathbf{W}_s$ in the RGB domain.

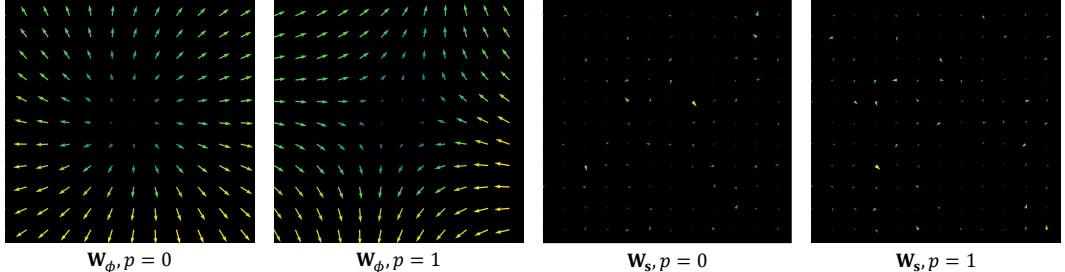

Figure 8: Visualization of the parametric embedding weight $\mathbf{W}_\phi$ and the standard embedding weight $\mathbf{W}_s$ in the vector field.

