# OpenReview forum: "Event2Vec: Processing neuromorphic events directly by representations in vector space"
_ICLR.cc/2026/Conference — ICLR 2026 Conference Withdrawn Submission_

### Official Review · Reviewer_fgEv · 2025-10-30

**Soundness:** 2
**Presentation:** 2
**Contribution:** 2
**Rating:** 4
**Confidence:** 4

**Summary:**

This paper introduces Event2Vec, a novel representation that encodes asynchronous events into vector space. By analogizing events to words in NLP, the authors propose spatial and temporal embedding modules that capture local structure and temporal dynamics, facilitating direct processing of event data using Transformer architectures.

**Strengths:**

1. The conceptual analogy between discrete language tokens and asynchronous neuromorphic events is both novel and compelling. It reframes event data not as irregular signals to be normalized, but as sequences to be understood in context.
2. The separation of spatial and temporal embeddings with inductive biases (e.g., spatial locality via parametric functions, temporal convolution on $\Delta t$) is methodologically solid.
3. Event2Vec achieves high throughput and minimal preprocessing time (especially with random sampling), showing practical relevance for real-time systems.

**Weaknesses:**

1. **Limited Task Diversity and Oversimplified Evaluation Benchmarks**:
The evaluation is confined to classification benchmarks. Given the strong NLP-inspired framing, it is disappointing that no sequence-centric assessments are included. Applications such as event-to-text generation, predictive modeling, or alignment with natural language would more fully test the paradigm. Moreover, the strong performance on simple datasets may thus reflect dataset limitations rather than model expressiveness. Additional experiments on more demanding benchmarks would be necessary to validate the model’s broader applicability.
2. **Preprocessing bottlenecks**:
While random sampling is fast, the K-Means variant—used for higher accuracy on DVS-Lip—incurs substantial latency. The trade-off undermines claims of efficiency, especially since no runtime adaptivity or approximations are proposed to mitigate the cost.
3. **Limited Robustness Analysis**:
The paper lacks evaluation under domain shifts (e.g., varying sensor resolutions, noisy events, different motion patterns). Given the sparse and sensitive nature of event data, robustness is critical.

**Questions:**

Please refer to the Weaknesses section.

**Details Of Ethics Concerns:**

No ethics concerns.

---

> ### Author Response · Authors · 2025-11-21
>
> > Limited Task Diversity and Oversimplified Evaluation Benchmarks
>
> In addition to the accuracy rate of classification tasks, our main text also provides a comprehensive evaluation, including parameter quantity, preprocessing time consumption, throughput, latency, and performance under the condition of a small number of events. In accordance with the comments from you and other reviewers, we have also supplemented more comparison items for your reference.
>
> > Preprocessing bottlenecks. … no runtime adaptivity or approximations are proposed to mitigate the cost
>
> Thank you for your suggestions. We have updated the latency comparison (including preprocessing) of different methods in "To all reviewers", and the results show that our method achieves lower latency. Additionally, we have adopted early stopping and approximation algorithms for k-means, which further reduces the latency on DVS Lip.

---

> ### Author Response · Authors · 2025-11-21
>
> > Limited Robustness Analysis
>
> We fully agree with your suggestions. As stated in our previous response, we do not consider accuracy to be the sole evaluation metric—other assessments are equally (if not more) important. We reproduced the `SNN + Frame (Yao et al., 2023)` network using its open-source code and compared its robustness with ours. The main reasons for selecting this model are as follows: first, the authors have made the complete training code publicly available; second, it achieved the state-of-the-art performance on the DVS Gesture dataset; third, SNNs are sometimes regarded as more robust due to their discrete spiking process [1-3]. It is worth noting that the accuracy we obtained through reproduction is 96.875, which is slightly lower than the performance reported in the original paper. We have contacted the original authors, who indicated that such slight discrepancies are normal due to inherent randomness in the training process.
>
> **Robustness of varying sensor resolutions**
>
> SNN: Bilinear interpolation is adopted to downscale the frame to a lower resolution, followed by upscaling back to the original resolution.
>
> Event2vec: The `x, y` coordinates are treated as floating-point numbers, scaled to the target resolution `(h, w),` and then quantized. Subsequently, the coordinates are upscaled back to the original resolution `(H, W)` and re-quantized.
>
> **Table R8: Comparison of robustness of resolution**
>
> | Resolution | Accuracy %(SNN \+ Frame) | Accuracy %(Event2vec) |
> | ---------- | ------------------------ | --------------------- |
> | 128x128    | 96.875                   | 97.57                 |
> | 96x96      | 93.750                   | 97.57                 |
> | 64x64      | 90.972                   | 97.60                 |
> | 48x48      | 92.014                   | 97.53                 |
> | 32x32      | 89.236                   | 97.63                 |
> | 16x16      | 81.597                   | 97.53                 |
> | 8x8        | 61.458                   | 96.70                 |
> | 4x4        | 30.903                   | 78.51                 |
> | 2x2        | 15.278                   | 26.25                 |
> | 1x1        | 5.208                    | 18.89                 |
>
> The results show that the event2vec method exhibits remarkable robustness to resolution. Even when the resolution is reduced to `1x1` (resulting in a complete loss of spatial information), event2vec can still achieve an accuracy of 18.89% using only temporal information.
>
> We perform random speed transformation on events to simulate motion patterns with varying speeds during shooting. Specifically, for an event stream containing L events, we randomly select `n_chunk` sub-event streams; the length of each sub-event stream is a random number between `[0, max_mask_ratio * L]`. For each sub-event stream, we multiply the time intervals between its events by a random number within `[scale_low, scale_high]`.
>
> **Table R9: Comparison of robustness of motions**
>
> | n\_chunk | max\_mask\_ratio % | scale\_low,scale\_high | Accuracy %(SNN \+ Frame) | Accuracy %(Event2vec) |
> | -------- | ------------------ | ---------------------- | ------------------------ | --------------------- |
> | 5        | 20                 | 0.5, 2                 | 94.79                    | 97.60                 |
> | 5        | 20                 | 0.25, 4                | 91.67                    | 96.60                 |
> | 5        | 20                 | 0.2, 5                 | 90.97                    | 93.89                 |
> | 10       | 20                 | 0.25, 4                | 88.19                    | 96.04                 |
> | 5        | 50                 | 0.25, 4                | 91.67                    | 96.18                 |
> | 5        | 100                | 0.25, 4                | 88.19                    | 96.22                 |
>
> The results demonstrate that event2vec also exhibits strong robustness to variations in motion patterns.
>
> \[1\] Sharmin, Saima, et al. "Inherent adversarial robustness of deep spiking neural networks: Effects of discrete input encoding and non-linear activations." European Conference on Computer Vision. Cham: Springer International Publishing, 2020\.
>
> \[2\] Sharmin, Saima, et al. "A comprehensive analysis on adversarial robustness of spiking neural networks." 2019 International Joint Conference on Neural Networks (IJCNN). IEEE, 2019\.
>
> [3] Ding, Jianhao, et al. ‘Neuromorphic Computing Paradigms Enhance Robustness through Spiking Neural Networks’. Nature Communications, vol. 16, no. 1, Nov. 2025, p. 10175.

---

### Official Review · Reviewer_E7eu · 2025-10-31

**Soundness:** 2
**Presentation:** 3
**Contribution:** 2
**Rating:** 4
**Confidence:** 4

**Summary:**

Inspired by word embeddings in NLP tasks, the authors designed event2vec to convert event data into vector representations, thereby introducing a novel way to represent event streams.
The proposed method embeds the event stream from both temporal and spatial dimensions, fully exploiting the characteristics of event flows. In Transformer-based experiments, the authors demonstrated that event2vec improves performance across multiple datasets while also providing an analysis of latency and efficiency.

**Strengths:**

1. Efficiency and Parameter Compactness. Experimental results show that Event2Vec maintains high accuracy while keeping the number of model parameters extremely small.
2. Temporal Generalization Ability. By employing a convolutional time embedding based on the relative time difference Δt, the model achieves time translation invariance, enabling it to generalize effectively across temporal shifts.

**Weaknesses:**

1. Limited Cross-Dataset Generalization. The model still requires separate training or fine-tuning on different datasets (such as DVS Gesture, ASL-DVS, and DVS-Lip), indicating that its zero-shot transfer capability remains weak.
2. Sensitivity to Clustering or Sampling Strategies. Because event stream lengths vary greatly, Event2Vec still relies on sampling or clustering to control input length. However, inappropriate strategy choices may lead to information loss or performance instability.
3. Limited Generalization and Heavy Preprocessing Requirements. Event2Vec demands extensive preprocessing, making it less suitable for real-time event streams or DVS-based video data, where its adaptability and efficiency are significantly constrained.

**Questions:**

See the section of weakness

---

> ### Author Response · Authors · 2025-11-21
>
> > Limited Cross-Dataset Generalization
>
> Thanks for your suggestions. We tested its linear probing accuracy \[1\] to verify its cross-dataset generalization ability. Specifically, we used the event2vec module and the first 5 layers of linear attention from the DVS Lip model (because it has the largest number of parameters among existing models), with their parameters frozen. We then added a single-layer fully connected layer after them as the classification head, and trained this setup separately on the DVS Gesture and ASL DVS datasets (it should be noted that we only trained the classification head). The accuracy rates obtained were 90.21% and 73.47% respectively, which indicates that our model indeed possesses strong cross-dataset generalization ability.
>
> \[1\] Radford, Alec, et al. "Learning transferable visual models from natural language supervision." International conference on machine learning. PMLR, 2021\.
>
> > Sensitivity to Clustering or Sampling Strategies.
>
> For clustering, we adopt the most commonly used K-Means algorithm instead of the more complex DBSCAN. Moreover, no parameter tuning is required—we simply normalize x, y, and t to the range \[0, 1\]. Regarding random sampling, Figure 3 demonstrates that our method exhibits remarkable robustness, achieving high performance even with an extremely small number of events. Figure 4 further shows that our method, using the simplest random sampling, completely outperforms the elaborately designed sampling methods under the Voxel Grid representation. Furthermore, we have also added the performance of using different k-means variants in the "To all reviews" section. You can observe that when early stop or approximation methods are employed, the accuracy rate of event2vec only decreases slightly, while the latency is significantly reduced.
>
>
> Therefore, we believe that event2vec is actually highly robust to clustering and sampling methods.
>
> > Limited Generalization and Heavy Preprocessing Requirements.
>
> We only use clustering when processing complex datasets; additionally, even on DVS Lip, our method can still achieve a relatively high performance of 70% with random sampling. Table 2 shows that random sampling is the fastest among all preprocessing methods. Furthermore, we have updated the end-to-end latency of inference in "To all reviewers" for your reference. You can find that our latency is actually lower than other methods, even with clustering.

---

> > ### Comment · Reviewer_E7eu · 2025-11-28
> >
> > While I appreciate the authors’ response, the current evidence remains insufficient to substantiate their key claims. For cross-dataset generalization, the authors only provide linear probing results rather than true zero-shot transfer or direct cross-dataset evaluation, which does not convincingly demonstrate generalization. Regarding robustness to clustering and sampling strategies, the analysis is limited to a few controlled cases and lacks systematic validation under diverse event distributions or more complex tasks. As for preprocessing overhead, the discussion focuses on offline scenarios and does not adequately address the method’s applicability to real-time event streams. Given these limitations, I believe the concerns remain unresolved and still constrain the strength of the work. Therefore, I will maintain my original score unless more comprehensive and convincing evidence can be provided.

---

> > > ### Author Response · Authors · 2025-12-02
> > >
> > > **Cross-Dataset Generalization Ability**
> > >
> > > Zero-shot transfer requires cross-modal alignment (e.g., event-text), yet mainstream event datasets do not provide text annotations. Direct cross-dataset evaluation is also infeasible, as the three datasets used in this paper have completely inconsistent class counts (11, 24, and 100). We argue that linear probing accuracy is widely adopted and recognized by researchers, which sufficiently demonstrates the cross-dataset generalization ability of our method.
> > >
> > > **Robustness to Clustering and Sampling Strategies**
> > >
> > > Our method achieves high performance with direct random sampling on all three datasets—attaining SOTA on ASL-DVS and outperforming the elaborately designed sampling strategies of traditional methods on DVS Gesture (Figure 4). Table R5 shows that even with poor clustering results, our method still achieves excellent performance on DVS Lip.
> > > It is worth noting that these three datasets are widely used and recognized by existing researchers, and their results are representative. Random sampling requires no hyperparameters, and the clustering algorithm also uses no hyperparameter tuning. Furthermore, we did not perform any denoising or other processing on the datasets. Therefore, we respectfully disagree with your classification of our experiments as "a few controlled cases."
> > >
> > > **Preprocessing Overhead**
> > >
> > > The current test results of our method (which you refer to as the offline scenario) are no different from real-time event stream processing, as our method needs to collect all event streams first before processing. It should be emphasized that the previous SOTA methods used for comparison in Table 1 follow the same logic:
> > >
> > > - SNN + Frame: Event integration requires division based on the length of the entire event stream; in addition, attention SNN does not have a causal structure and needs to use data from all time steps simultaneously.
> > > - GNN, Transformer + Image, Voxel Graph: Image preprocessing is consistent with the above; They use an encoder-only Transformer which requires data from all time steps.
> > > - Spiking ResNet18, BiGRU + Frame: Frame preprocessing is consistent with the above; bidirectional GRU is used, which requires data from all time steps.

---

### Official Review · Reviewer_EEdR · 2025-11-01

**Soundness:** 3
**Presentation:** 3
**Contribution:** 2
**Rating:** 4
**Confidence:** 3

**Summary:**

To address the incompatibility between the asynchronous and sparse event streams output by event cameras and mainstream deep learning paradigms, this paper proposes a representation method named Event2Vec. It draws an analogy between events and words in natural language, enabling direct processing of events by embedding them into a vector space. The core of the method includes parametric spatial embedding (which incorporates neighborhood similarity bias) and convolutional temporal embedding based on temporal differences, supporting random sampling or k-means clustering sampling to fix the sequence length. Experiments on the DVS Gesture, ASL-DVS, and DVS-Lip datasets verify its advantages: high parameter efficiency, fast preprocessing, high throughput, and robustness with an extremely low number of events. Its accuracy is comparable to that of representative methods. The authors emphasize that this method provides a new paradigm for event stream processing, facilitating integration with Transformers and self-supervised learning, and paving the way for multimodal fusion.

**Strengths:**

The paper introduces a novel concept and clearly explains its motivation through an analogy between events and words. The spatial embedding employs a parametric network that explicitly injects neighborhood similarity bias, and the visualizations demonstrate smooth gradients. For temporal modeling, it uses differential convolution, which ensures time-shift invariance and local contextual consistency. From an engineering perspective, the approach offers extremely low preprocessing latency and high training and inference throughput, significantly outperforming voxel and point cloud-based methods. It also maintains reasonable accuracy even with very few events, showing strong robustness. In addition, the model supports self-supervised pre-training, and its performance further improves after fine-tuning on DVS-Lip, demonstrating good extensibility.

**Weaknesses:**

The paper lacks sufficient comparisons with the strongest baselines under unified settings. For example, although the 99.91% accuracy on ASL-DVS is high, the accuracy on DVS-Lip only reaches 75.14% through clustering and pre-training, which is lower than the 75.3% achieved by the SNN + Frame method. There is no fair ablation study conducted under the same backbone and resource conditions. The k-means clustering sampling process also introduces high preprocessing time, which may limit its applicability in online scenarios. The temporal modeling relies solely on first-order differential convolution, and the verification for long-range dependencies remains insufficient. The task evaluation is primarily focused on classification, without evidence of effectiveness in multi-task scenarios such as detection or reconstruction. The ablation experiments are limited in scope and fail to thoroughly examine the robustness of the intensity factor 𝜌, attention variants, or sampling strategies. Moreover, the transparency of reproducibility details, including training time and sources of variance, should be improved.

**Questions:**

Does parameter sharing limit expressiveness, and what are the comparison results with bidirectional models that do not use parameter sharing?

What is the sensitivity of the intensity factor $\(\rho\)$ to distribution, and what impact would it have if $\(\rho\)$ is treated as an additional channel or processed through normalization?

In real-time applications, what are the specific end-to-end latency and throughput of clustering and random sampling, and is it possible to integrate clustering into a trainable module?

Can Event2Vec achieve zero-shot transfer to tasks such as optical flow or depth estimation, or align with large language models (LLMs) to implement multimodal tasks?

Why is data augmentation not used for ASL-DVS while multiple data augmentations are applied for DVS Gesture, and does this affect the fairness of comparison?

---

> ### Author Response · Authors · 2025-11-21
>
> > The paper lacks sufficient comparisons with the strongest baselines under unified settings. There is no fair ablation study conducted under the same backbone and resource conditions.
>
> Our method is event-wise and processes the event stream as a sequence similar to natural language using linear attention. This is completely different from the traditional approaches that rely on either "dense representation \+ CNN/SNN" or "sparse representation \+ sparse network". Therefore, a direct comparison under the same network structure as SNNs cannot be conducted. However, we provide a comprehensive set of metrics, including accuracy rate, number of parameters, preprocessing time, throughput, and inference latency, enabling a full-scale evaluation of different types of methods.
>
> We trained using the original code of `Spiking ResNet18 and BiGRU + Frame` on DVS Lip, and compared the training cost of our model in the same environment (NVIDIA A100 80GB \+ 8 CPU cores \+ 256G RAM) for your reference:
>
> **Table R6: Comparison of training costs on DVS Lip**
>
> |                                  | Spiking ResNet18 and BiGRU \+ Frame | Event2vec                                                |
> | -------------------------------- | ----------------------------------- | -------------------------------------------------------- |
> | GPU Memory (MiB)                 | 25113                               | 10593 (Self-supervised training), 10311 (Classification) |
> | GPU Hours                        | 97.73                               | 6.15                                                     |
> | Training throughput (samples/s)  | 8.65                                | 281.80                                                   |
> | Inference throughput (samples/s) | 78.79                               | 829.80                                                   |
>
> You can see that our method only uses 41% of the memory consumption and 6.3% of the GPU training time of their method.
>
> > The k-means clustering sampling process also introduces high preprocessing time, which may limit its applicability in online scenarios.
>
> K-Means clustering does significantly increase latency, but this only occurs when it is used on the complex DVS Lip dataset. For DVS Gesture and ASL DVS, our method achieves sufficiently high performance using only random sampling. It is worth noting that even when random sampling is used on DVS Lip, our method can still reach a performance of 70.62%. Please also refer to the supplementary experimental results in our "To all reviewers". You will find that our end-to-end latency on DVS Lip is actually lower than `Spiking ResNet18 and BiGRU + Frame`.
>
> > The task evaluation is primarily focused on classification, without evidence of effectiveness in multi-task scenarios such as detection or reconstruction.
>
> Thank you for your suggestions. Detection and reconstruction rely on dense representations, which is indeed somewhat challenging for event-wise representations, as they are essentially sparse representations. In Table 3 of the original FARSE-CNN paper (Santambrogio et al., 2024), the object detection performance of several sparse representation methods on Gen1 is compared. It can be seen that most methods achieve extremely low performance, with only FARSE-CNN obtaining an mAP of 30%. Notably, the features output by FARSE-CNN for detection are also dense; subsequently, these features are converted into dense feature maps using `scatter` and then used for object detection with YOLO.
>
> We conducted tests using a similar approach: on the Gen1 dataset, we randomly sampled 65,536 events, scattered the output features of event2vec, and performed detection using DINO \[1\]. This approach achieved an mAP of 32.94 ± 0.06%, which is higher than that of FARSE-CNN. However, we believe this method is not elegant. An ideal event-wise object detection should be performed directly on sparse features, and preferably directly on sequences: the detection boxes would no longer be 2D but 3D (the additional dimension is time). This is an exciting new research topic, as current event-based object detection methods are all based on a fixed frequency, focusing only on detecting "where" rather than "when and where". Nevertheless, the workload required for this research exceeds the scope of this paper.
>
> \[1\] Zhang, Hao, et al. "DINO: DETR with Improved DeNoising Anchor Boxes for End-to-End Object Detection." The Eleventh International Conference on Learning Representations.

---

> > ### Author Response · Authors · 2025-11-21
> >
> > > Moreover, the transparency of reproducibility details, including training time and sources of variance, should be improved.
> >
> > Thank you for your suggestion. As a matter of fact, the source code we provided includes the original TensorBoard logs (e.g., `logs_/asl_dvs/events.out.tfevents.1756509791.qwe-SYS-420GP-TNR.2466019.0`) and command-line outputs (e.g., `logs_/asl_dvs/out.txt`), from which you can easily check the training time. For example, you can check `logs_/tb.png` we provided in code and find the total training time for dvs\_lip is “46m 10s”.
> >
> > We performed sampling on the test set 10 times, then calculated the mean and standard deviation of the 10 results. This is a fully automated process that has been integrated into the code—readers can directly reproduce the results by running the training process or loading the pre-trained model we provided.
> >
> > > Does parameter sharing limit expressiveness, and what are the comparison results with bidirectional models that do not use parameter sharing?
> >
> > Thank you for your suggestion. We explored this aspect in our early experiments. Generally speaking, not sharing parameters can indeed enhance the fitting capability; however, training on event data is highly prone to overfitting. Therefore, the reduction in fitting capability caused by parameter sharing is not necessarily detrimental. Below are our experimental results on three datasets:
> >
> > **Table R7: Accuracy change with no-shared parameters in bidirectional attentions**
> >
> > | Dataset     | Test accuracy (%) | Params (MB)  |
> > | ----------- | ----------------- | ------------ |
> > | DVS Gesture | 96.63 (-0.94)     | 0.65 (+25%)  |
> > | ASL DVS     | 99.86 (-0.05)     | 0.34 (+26%)  |
> > | DVS Lip     | 74.10 (-1.04)     | 22.90 (+25%) |
> >
> > > What is the sensitivity of the intensity factor to distribution, and what impact would it have if is treated as an additional channel or processed through normalization?
> >
> > Thank you for your suggestion. We have calculated the statistics (maximum, mean, and standard deviation) of the intensity for individual samples across the entire training set, and the results are as follows:
> >
> > maximum: 49.41 ± 26.15
> >
> > mean: 10.66 ± 7.53
> >
> > std: 8.95 ± 5.28
> >
> > From the statistical results, it is evident that there are significant differences in the intensity distribution among different samples. Following your suggestion, we attempted to use intensity as an additional channel, inputting it together with x, y, and p. However, the network failed to converge in this case.
> >
> > We then normalized the intensity by `intensity = log(intensity) + 1` and found that the performance on the DVS Lip dataset improved slightly, reaching 75.38%. Therefore, your suggestion regarding optimizing the way intensity is utilized has been highly helpful.
> >
> > > In real-time applications, what are the specific end-to-end latency and throughput of clustering and random sampling, and is it possible to integrate clustering into a trainable module?
> >
> > Please refer to "To all reviewers" for results about end-to-end latency. The end-to-end throughput of clustering and running models on DVS Lip is about 9 samples/s for training and 11 samples/s for inference. This is extremely slow because the clustering is the bottleneck. Fortunately, we can always pre-culster once and train for many times.
> >
> > The throughputs of event2vec models we reported in this paper already include random sampling.
> >
> > We believe that clustering can be integrated into training, and \[2\] serves as an excellent example. They implemented flash-kmeans and used it for visual tokens. Overall, both event tokens and visual tokens (especially in video tasks) face the problem of an excessive number of tokens. We believe that research in these two fields can be integrated with each other to promote mutual development.
> >
> > \[2\] Yang, Shuo, et al. "Sparse VideoGen2: Accelerate Video Generation with Sparse Attention via Semantic-Aware Permutation." The Thirty-ninth Annual Conference on Neural Information Processing Systems

---

> > > ### Author Response · Authors · 2025-11-21
> > >
> > > > Can Event2Vec achieve zero-shot transfer to tasks such as optical flow or depth estimation, or align with large language models (LLMs) to implement multimodal tasks?
> > >
> > > Regarding the transfer learning capability, you may refer to our response to Reviewer E7eu’s comment on "Limited Cross-Dataset Generalization".
> > >
> > > Regarding optical flow and depth estimation, we believe the current main challenges are similar to those faced by object detection as discussed with you earlier—all these tasks rely on dense representations. However, by leveraging scatter operations to enable conversion between sparse and dense representations, we argue that event2vec can at least accomplish these tasks by integrating traditional methods: it only requires replacing the network input with the feature map (shape \= \[D, H, W\]) obtained through scatter operations on event tokens (shape \= \[L, D\]).
> > >
> > > One of the main issues currently plaguing multimodal Large Language Models (LLMs) is how to align tokenizers across different domains. We believe that event2vec provides an elegant solution for event tokenization, and thus, it is entirely possible in the future to obtain a foundational large model through large-scale self-supervised training, which can then be fine-tuned to accomplish the tasks you mentioned.
> > >
> > > > Why is data augmentation not used for ASL-DVS while multiple data augmentations are applied for DVS Gesture, and does this affect the fairness of comparison?
> > >
> > > Please refer to “To All reviewers”. In general, all methods are equipped with data augmentations, and the comparison is fair.

---

### Official Review · Reviewer_4MEw · 2025-11-01

**Soundness:** 1
**Presentation:** 2
**Contribution:** 2
**Rating:** 2
**Confidence:** 4

**Summary:**

The paper proposes event-2-vec for event-based cameras that convert each event into vectorized representations, added with positional encoding terms from the temporal side of the input, using temporal differences. Motivated from NLP based inputs to transformers which are vectorized with added positional encoding, the authors follow a similar structure here. The resulting approach is more parameter-efficient and experiments across three datasets show decent performance. Attention maps are shown to highlight the explainability of the method.

**Strengths:**

1. Paper is written well overall
2. Attention maps of their approach are highlighted, which adds good explainability of the proposed approach.
3. Parameter efficiency is good.

**Weaknesses:**

Ambiguous use of words to describe the datasets and some of the stats (single stream vs samples; throughput etc.) Proper definitions would be good for ease of readability. Otherwise the reader will need to infer things themselves, as I had to. There are still some questions at large.

Ultimately what matters is single-stream latency, as in the wild a camera won't be processing 1000 sequences at once when live, but just one. From that perspective, due to the transformer's bulk and large input context, I see that the authors' approach is actually slower than others except the FARSE-CNN approach. This significantly changes the narrative. The parameter efficiency is not useful if the actual deployment latencies are larger than other approaches, while the performance being worse (e.g. SNN+ Frame exhibits almost half latency while being more accurate on DVS-Gesture). Also, it is not clear if the single-stream latency also includes the pre-processing of those events within the stream. Even if it does not, I note that the results in Table 2 essentially suggest that the pre-processing time for the first four approaches are negligible, as these are "total" times across 1000+ total samples, which significantly reduces the pre-processing time per-sample.

I think the temporal embedding here is a bit of an issue. I feel the authors' tried a bunch of encodings (which is demonstrated in Tab. 4) and the narrative seems to have been shaped around what empirically worked. I was not convinced by 3.3, as it seemed that due to the fact that time can be open-ended (and thus the risk of "out-of-distribution") to ensure it is not, the authors picked some range limiting approaches and went with what worked best. Their final choice also does not align well with the original NLP based motivation (lines 71-92). In the transformer context a "position encoding" is usually given to the information corresponding to a input's position, and usually uses sin/cosine embeddings or even learnable ones. So the natural choice for PEs in this context would also be sin and cosine (as the authors have indeed explored). Lines 71-92 motivates either a direct normalized map of times or a function imposed on them (sin/cosine). However, the authors go with a function of the time difference of consecutive events, and for me the motivation for this is lacking. Firstly, having worked in this space, I'm aware of the quite random nature of the arrival of events, and I'm not sure how interpretable or sensible the consecutive temporal difference of two events would be, as the events themselves could come from very different parts of the input. I suppose what is happening is that in such datasets (like DVS-Gesture), the temporal difference embedding is essentially just capturing the rate at which the events are happening, which causally links to the current "speed" of the motion. Faster motion will simply yield (on average) lower $\Delta t$ and vice-versa. That to me makes sense more than what the authors' outline in 3.3, which is temporal shift invariance and the role of convolutions. I also do not see the actual utility of using a convolutional encoder, as the authors seem to be working with a single $\Delta t$ input to the conv-encoder in Figure 2 anyway (please correct me if I'm wrong). Either way, I don't see the proper motivation behind using a convolutional encoder versus any other from the authors' motivations. The only way it makes sense to me is perhaps from the fact that conv architectures are more parameter efficient. The design choices in Figure 2 appear somewhat ad-hoc as well.


Following up on the previous point, this leads to one of the core issues for me in this work. The authors' main motivation is to avoid dense event-to-frame representations which can lead to "degradation or complete loss of high temporal resolution" (line 124). However, given the relatively noisy nature of these cameras in capturing the events, and given the random nature of the consecutive temporal event differences other than just giving a rough indication of the degree of motion overall in the scene, it is not clear to me how the framework resolves the aforementioned issues without creating these new ones. Excessive dependence on temporal information with event-based cameras can be counter-productive, and that is why framing/aggregating has happened in the first place with most works in literature. Lastly, I note that even in this work, where not sacrificing temporal information is key, the authors inevitably end up using an aggregation approach. The motivation is cited as maintaining the fixed context length L, but I believe aggregation also yields the necessary invariance to noisy local temporal information. The clustering approach mentioned here essentially does that, while the random sampling approach seems to lose information as it is subsampling the events.


I note that the authors indeed follow a very rigorous data augmentation procedure for all datasets, however, I don't see any references for the set of augmentations used, so it is not clear whether accuracy wise all the quoted methods are being compared on equal grounds, as data augmentation can significantly enhance accuracy.


I feel the experiments would benefit from more baselines, as DVS-Gesture is one of the oldest datasets and there are many approaches out there. A survey from 2023 outlines higher sota accuracy (https://ieeexplore.ieee.org/stamp/stamp.jsp?tp=&arnumber=10298106) than the methods discussed in the paper.

**Questions:**

Please see weaknesses.

---

> ### Author Response · Authors · 2025-11-21
>
> It has been a pleasure communicating with you. Many of your ideas align with ours in the design of event2vec, and we will elaborate on our thinking regarding these questions below.
>
> > Ambiguous use of words to describe the datasets and some of the stats
>
> We are sorry for that.  A "sample" means an individual sample in a dataset, which is also a "single event stream". For example, there are 1176 train samples in the DVS Gesture train set.
>
> "Throughput (samples/s)" is the number of batched samples the model can process in one second.
>
> "Single Stream Latency" is the time of the model to process a sample (a single event stream). It is the time difference between when we input a sample to the model and when we get the output. This metric in Table 3 is averaged in the test set during inference with `batch size = 1`.
>
> > I see that the authors' approach is actually slower than others except the FARSE-CNN approach
>
> We apologize for the lack of clarity in our writing that caused confusion. In fact, Table 3 does not include the preprocessing time. Please refer to the supplementary experimental results in our "To all reviewers". You will find that our end-to-end latency on both DVS Gesture and DVS Lip is lower than that of other methods.
>
> > It is not clear if the single-stream latency also includes the pre-processing of those events within the stream.
>
> The pre-processing time of event2vec is included in Table 3 because we use random sampling on the fly. But pre-processing times of other methods are not because these methods just load processed data. In fact, we can also save randomly sampled event streams in advance and load them in training. But it is not necessary because randomly sampling is fast enough (refer to Table 2).
>
> > I'm aware of the quite random nature of the arrival of events, and I'm not sure how interpretable or sensible the consecutive temporal difference of two events would be, as the events themselves could come from very different parts of the input.
>
> We fully agree with your concern. Fortunately, the information about "different parts of the input" is absorbed into event tokens by the spatial embedding. As Figure 6 shows, the trained linear attentions are able to distinguish the importance of events in different spatial positions.
>
> > I also do not see the actual utility of using a convolutional encoder, as the authors seem to be working with a single input to the conv-encoder in Figure 2 anyway (please correct me if I'm wrong)
>
> We are sorry for the misunderstanding caused by our writing. In fact, the inputs for the convolutional encoder are "a sequence of time differences", e.g., `{t[1] - t[0], t[2] - t[1], ..., t[L-1] - t[L-2], padding}`.
>
> > The design choices in Figure 2 appear somewhat ad-hoc as well.
>
> We are glad to give a more detailed explanation. The spatial embedding should process 3-dimensional inputs and output $D$-dimensional tensors. The multilayer perceptron (MLP) is a general choice. For temporal embedding, we want to incorporate additional contextual consistency (refer to Section 3.3). Thus, we use multilayer convolutions. Overall, our design adheres to Occam’s Razor: selecting the simplest network structure on the premise of meeting the requirements.
>
> > Why use the time difference
>
> First, no matter the convolutional layers or linear attentions, their operations can be regarded as the sum of inputs along the sequence length dimension. Thus, when we input $\\Delta t$, the sum of $\\Delta t$ actually becomes the integration of events. This idea is similar to previous event representations: for example, the frame representation just integrates events in a certain period. Second, $\\Delta t$ carries some important information, such as the speed of events (your view) and temporal intensity. Third, although conversion from $t$ to $\\Delta t$ only requires a difference operation, it may be hard for the network to learn and operate on $t$ directly. As the famous ResNet paper \[1\] points out, "It has been shown that these solvers (solve residual solutions) converge much faster than standard solvers that are unaware of the residual nature of the solutions. These methods suggest that a good reformulation or preconditioning can simplify the optimization".
>
>
> \[1\] He, Kaiming, et al. "Deep residual learning for image recognition." Proceedings of the IEEE conference on computer vision and pattern recognition. 2016\.

---

> > ### Author Response · Authors · 2025-11-21
> >
> > > However, given the relatively noisy nature of these cameras in capturing the events, and given the random nature of the consecutive temporal event differences other than just giving a rough indication of the degree of motion overall in the scene, it is not clear to me how the framework resolves the aforementioned issues without creating these new ones.
> >
> > Yes, the randomness and noise of events are a big problem. This is also the reason why we spend a lot of space discussing parametric spatial embedding. An event occurs at $(x,y)$ or $(x,y+1)$ has no much difference, but will be embedded to two different tokens by a standard embedding layer. Then we proposed the parametric spatial embedding method, and events with close coordinates are embedded into similar tokens.
> >
> > > Lastly, I note that even in this work, where not sacrificing temporal information is key, the authors inevitably end up using an aggregation approach. The motivation is cited as maintaining the fixed context length L, but I believe aggregation also yields the necessary invariance to noisy local temporal information. The clustering approach mentioned here essentially does that, while the random sampling approach seems to lose information as it is subsampling the events.
> >
> > As you mentioned, random sampling leads to the loss of a large amount of information. Therefore, when dealing with a relatively challenging dataset like DVS Lip, we adopted a clustering method. By adding intensity as an input, we ensured that all events could contribute to the final output. Clustering also brings additional benefits, such as noise removal, as you pointed out. We highly agree with your viewpoint on event noise and believe that the optimal solution to this problem is to increase the number of input events, for example, to tens of thousands. However, our experiments showed that this approach did not yield good results. The primary reason is likely that linear attention is still an RNN, which has a limited memory capacity and cannot maintain long-term memory without degradation when processing such long input sequences.
> >
> > > I don't see any references for the set of augmentations used
> >
> > Please refer to "To all reviewers".
> >
> > > I feel the experiments would benefit from more baselines. A survey from 2023 outlines higher sota accuracy.
> >
> > Thanks for your suggestions, and we will add more baselines. It is true that many methods have achieved better performance than the one proposed in this paper on the DVS Gesture dataset. However, we believe that performance is only one of the evaluation metrics, and our method has a distinct advantage in terms of the number of parameters. Additionally, since the DVS Gesture test set contains only 288 samples, there is actually a difference of merely 2 correctly classified samples between our method and the Spiking Neural Network (SNN) with the highest performance in Table 1\. The article you provided achieved a 99.5% accuracy rate on the DVS Gesture dataset; yet, given the test set’s size of 288 samples, an accuracy rate of 99.65% should be achieved if 287 samples are classified correctly, and 99.30% if 286 samples are classified correctly. Moreover, no evidence of the authors using the average result of multiple tests was found in either the original paper or the open-source code. Therefore, we suspect there might be minor issues with the authors’ experiments (for instance, when I served as a reviewer previously, I encountered a case where an author claimed to have achieved 100% accuracy on DVS Gesture, but upon checking their code, I discovered they had forgotten to set train=False in the test set code). In addition, the code provided in this article does not include training hyperparameters or the code for loading the DVS Gesture dataset, making it impossible for me to reproduce their results.
> >
> > > Why is data augmentation not used for ASL-DVS while multiple data augmentations are applied for DVS Gesture, and does this affect the fairness of comparison?
> >
> > Please refer to "To all reviewers".

---

### Author Response · Authors · 2025-11-21
**To all reviewers**

# Fix Table 1

In Table 1, we wrongly filled in the accuracy and parameters of `ResNet-18, BiGRU + Frame`. And the params (MB) of the first two rows should multiply  4, because the original data from Table 1 in (Dampfhoffer & Mesquida, 2024\) are `the number of params` rather than `params`. The relations between them are: `params (MB) = the number of params (MB) * 4` because a float32 value occupies 4 bytes. Thus, our method is only 0.16% lower than theirs, but it only uses 8.18% params of their. After fixing the errors, the parameter efficiency of the method proposed in this paper has been further verified.

**Table R1: Comparison on the DVS Lip Dataset**

| Dataset | Method \+ Representation                                     | Accuracy (%)                             | Params (MB) |
| ------- | ------------------------------------------------------------ | ---------------------------------------- | ----------- |
| DVS-Lip | ResNet-18,BiGRU \+ Frame (Tan et al., 2022\)                 | 72.1                                     | 241.2       |
|         | Spiking ResNet18,BiGRU \+ Frame (Dampfhoffer & Mesquida, 2024\) | 75.3                                     | 223.63      |
|         | Linear Attention \+ Event2vec (1024 Randomly Sampled Events) | 70.62±1.55                               | 18.3        |
|         | Linear Attention \+ Event2vec (1024 Cluster Events)          | 75.14, 75.38 (with normalized intensity) | 18.3        |

In addition, as suggested by Reviewer EEdR, we normalize the intensity in accordance, and our new model achieves an accuracy rate of 75.38%, outperforming the performance of `Spiking ResNet18,BiGRU + Frame (Dampfhoffer & Mesquida, 2024)`.  We also compare the training costs and find that our method only uses 41% of the memory consumption and 6.3% of the GPU training time of their method.

# Fair comparison with data augmentation

Transformers (including linear attention) lack inductive bias and thus require more data for learning. We used data augmentation methods to expand the data volume to a certain extent, thereby improving performance. Specifically, we did not use data augmentation on the ASL-DVS dataset because we found that SOTA (State-of-the-Art) performance could be achieved without it—this is likely due to the sufficient scale of this dataset: the number of samples in its training set is approximately 80,640, while that of DVS Gesture is 1,176, and DVS Lip is 14,896.

We clearly provide all the details of data augmentation, as well as complete code, hyperparameter configuration files, and training logs to facilitate readers' reproduction. Most of the previous studies also used data augmentation. Since we have not proposed a plug-and-play module, such as a new network layer or learning algorithm, but rather a brand-new framework, we believe it is reasonable to use data augmentation and compare it with existing SOTA (State-of-the-Art) studies.

The following Table R2 are the data augmentation methods used in the articles in Table 1\. It can be observed that all existing methods utilize data augmentation.

**Table R2: Data augmentations of methods in Table 1**

| Dataset     | Method \+ Representation             | Data Augmentation                                            |
| ----------- | ------------------------------------ | ------------------------------------------------------------ |
| DVS Gesture | DVS Gesture Sparse GRU \+ Frame      | Random crop, translation, and rotation                       |
|             | SNN \+ Frame                         | Random slice and integrate                                   |
|             | FARSE-CNN \+ Window Slicing          | Random coordinate translations                               |
|             | Event MAE \+ Point Cloud             | Point resampling from Point-BERT (CVPR 2022\)                |
|             | Linear Attention \+ Event2vec        | Random resize, rotation, shear, translate, erase, and chunk dropout |
| ASL-DVS     | GNN,CNN \+ Graph                     | Random scale, flip, and rotation of  node positions          |
|             | GNN,Transformer \+ Image,Voxel Graph | Random scale and translate                                   |
|             | Linear Attention \+ Event2vec | None                                                         |
| DVS-Lip     | ResNet-18,BiGRU \+ Frame             | Random crop and horizontal flip                              |
|             | Spiking ResNet18,BiGRU \+ Frame      | Random crop, horizontal flip, spatial masking, zoom, and temporal mask |
|             | Linear Attention \+ Event2vec        | Random resize, rotate, shear, flip, translate, and erase     |

---

> ### Author Response · Authors · 2025-11-21
>
> # Latency with pre-processing
>
> It should be noted that Table 3 in the original paper does not include the preprocessing latency. In addition, the time consumption of SNN \+ Frame and Sparse GRU \+ Frame in Table 3 was wrongly underestimated, and the reasons will be elaborated on in the following text. Table R3 shows the end-to-end latency, including both pre-processing and inference times, of single event stream inference latency on DVS Gesture. The random sampling of event2vec requires no preprocessing, so the latency data remains consistent with the original. It can be found that when using random sampling, our method achieves the lowest latency. It should be noted that event2vec does not actually require the use of K-Means on the DVS Gesture dataset—this method is only employed on DVS Lip.
>
> **Table R3: End-to-end latency of single event stream inference latency on DVS Gesture**
>
> | Method                         | End-to-end latency (Pre-processing+inference) (ms) |
> | ------------------------------ | -------------------------------------------------- |
> | FARSE-CNN \+ Window Slicing    | 598.76                                             |
> | AEGNN \+ Graph                 | 56.21                                              |
> | SNN \+ Frame                   | 191.28                                             |
> | Sparse GRU \+ Frame            | 175.70                                             |
> | Event2vec \+ Randomly Sampling | 11.44                                              |
>
> Note:
>
> SNN+Frame: In the original paper, we reproduced SNN+Frame using SpikingJelly (refer to Table 6 in the appendix) with 16 frames. This is a fairly common setup in SNNs. It should be noted that this setup fails to achieve the original performance reported in (Yao et al., 2023):  following the suggestion of Reviewer fgEv, we conducted robustness experiments and found that the accuracy of our reproduction was significantly lower than that of the original open-source code. It should be noted that the original work actually used 60 frames (refer to this link: [https://github.com/BICLab/Attention-SNN/issues/4\#issuecomment-2080249319](https://github.com/BICLab/Attention-SNN/issues/4#issuecomment-2080249319) ) and did not use SpikingJelly. We have now switched to the original code for experiments, and thus the latency data in the above table has been updated accordingly. Specifically, the inference latency per sample of their original code is approximately 172.41 ms, and the data preprocessing time is about 18.87 ms. We will fix the relevant content in the next version of the paper.
>
> Sparse GRU \+ Frame: The slow speed of Sparse GRU \+ Frame is due to the fact that this method splits one sample into 3.5 new samples on average, so the total time consumption needs to be multiplied by 3.5. However, we forgot to apply this 3.5x multiplier in Table 3 of the original paper (We did not identify this issue at that time).
>
> Furthermore, we compared the latency on the DVS Lip dataset with `Spiking ResNet18 and BiGRU + Frame` in Table R4:
>
> **Table R4: End-to-end latency of single event stream inference latency on DVS Lip**
>
> | Method                              | End-to-end latency (Pre-processing+inference) (ms) |
> | ----------------------------------- | -------------------------------------------------- |
> | Spiking ResNet18 and BiGRU \+ Frame | 362.04                                             |
> | Event2vec \+ Random Sampling        | 36.03                                              |
> | Event2vec \+ K-Means                | 88.58                                              |
>
> The results show that even with the use of K-Means, event2vec only increases the latency by approximately 50 ms, and the total latency remains significantly lower than that of `Spiking ResNet18 and BiGRU + Frame`.
>
> According to the suggestions from reviewer fgEv, we have also provided the results of some early stop methods and fast algorithms for k-means in Table R5:
>
> **Table R5: Accuracy on DVS Lip of using variants of K-Means**
>
> | Method                                                       | Latency (ms) | Accuracy (%) |
> | :----------------------------------------------------------- | :----------- | :----------- |
> | K-Means                                                      | 88.58        | 75.14        |
> | K-Means with early stop when relative tolerance of iterations \< 0.1 | 75.87        | 75.12        |
> | K-Means with only one iteration                              | 69.63        | 74.87        |
> | Mini-batch K-Means (batch size \= 1024, maximum iterations \= 100\) | 53.13        | 73.43        |
> | Random sampling                                              | 38.62        | 70.62        |
>
> The above results show that in practical applications, we can make flexible choices based on the trade-off between latency and accuracy.

---

### Author Response · Authors · 2025-12-02
**Summary of Reviews and Responses**

To facilitate the AC's review, we have summarized the reviewers' comments and our responses below.

**Strengths**

- All reviewers recognized the extremely high parameter efficiency of the proposed method and its excellent performance on three datasets. Table 1, after our revisions, further illustrates the parameter efficiency (see Table R1 in the rebuttal).
- Reviewer 4MEw praised the visualization of attention in this paper, believing it highlights the interpretability of the proposed method.
- Reviewers EEdR and fgEv acknowledged the motivation and novelty of the paper, stating that drawing inspiration from natural language processing to understand event data through context is reasonable.
- Reviewers EEdR and E7eu commended the proposed method for maintaining high accuracy even with an extremely small number of events.
- Reviewers EEdR and fgEv praised the method for its extremely high throughput, which is of great significance for real-time systems.

**Issues**

- Reviewers 4MEw and EEdR noticed that we used extensive data augmentation and questioned whether this would lead to unfair comparison. We investigated other methods used for comparison and confirmed that all methods employed data augmentation (Table R2), thus ensuring the fairness of the comparison.
- Reviewers 4MEw, EEdR, and fgEv all focused on the end-to-end latency of the proposed method, including data processing—especially whether the k-means algorithm used in the DVS Lip dataset would significantly increase latency. By comparing with existing SOTA methods, we found that even with the k-means algorithm, the latency of our method is much lower than the previous SOTA method (Table R4). For the DVS Gesture dataset, our method only requires random sampling, resulting in latency much lower than that of other methods (Table R3).

**Controversies**

- Reviewer 4MEw argued that the differential encoding of timestamps in our method is unreasonable.
- However, other reviewers regarded this method as one of the significant advantages of the paper. For instance, Reviewer EEdR stated that this method "ensures time-shift invariance and local contextual consistency"; Reviewer E7eu noted that it "achieves time translation invariance, enabling it to generalize effectively across temporal shifts"; and Reviewer fgEv considered it "methodologically solid".
- We provided a detailed explanation in the separate response to Reviewer 4MEw, clarifying that this design first performs differentiation on timestamps, then conducts integration via convolution and linear attention. This achieves an effect similar to residual learning and is easier to optimize than directly learning from the timestamps themselves.

**Supplementary Experiments**

- Reviewer EEdR put forward some innovative experimental suggestions, which we attempted to implement. Although most of these methods yielded poor results (some of which we had already tried before), the idea of normalizing cluster intensity proposed by the reviewer achieved good results. This further improved the performance of our method on the DVS Lip dataset, surpassing the previous SOTA method.
- Reviewer EEdR inquired about the performance of our method in detection or reconstruction tasks. We responded truthfully that current methods based on sparse representation struggle to handle these tasks. However, after adopting a sparse-to-dense conversion method similar to that in the FARSE-CNN paper, our method outperformed FARSE-CNN (as well as other sparse methods that originally had lower performance than FARSE-CNN) in object detection on the Gen1 dataset.
- Reviewer E7eu asked about the cross-dataset generalization ability of our method. We tested the model's linear probing accuracy and found that it possesses good cross-dataset generalization ability.
- Reviewer E7eu also inquired about the robustness of our method to clustering approaches. The results in Table R5 show that our method is not sensitive to clustering effects and can make flexible choices based on the trade-off between latency and accuracy.
- Reviewer fgEv asked about the robustness of our method. The results in Tables R8 and R9 demonstrate that our method has extremely strong robustness.

---

### Note · Authors · 2025-12-15

**Comment:**

We recently identified a critical bug during code consolidation, which resulted in the latency values reported in the main manuscript and rebuttal being inaccurately low. Specifically, while we set `batch_size=1` for single-sample inference, we discovered that the default configuration of `number_workers=8` caused the data loader to preload and process data in the background—an oversight that invalidates the reported latency measurements.

These erroneous results risk misleading the reviewers and AC, and we sincerely apologize for this mistake. Given the impact on the validity of our findings, we formally withdraw our submission.

**Withdrawal Confirmation:**

I have read and agree with the venue's withdrawal policy on behalf of myself and my co-authors.